# An Error Reduction Algorithm to Improve Lidar Turbulence Estimates for Wind Energy

Jennifer F. Newman[1] and Andrew Clifton[2]

[1]National Wind Technology Center, National Renewable Energy Laboratory, Golden, CO, 80401, USA
[2]Power Systems Engineering Center, National Renewable Energy Laboratory, Golden, CO, 80401, USA

*Correspondence to:* Jennifer F. Newman (Jennifer.Newman@nrel.gov)

**Abstract.**

Remote sensing devices such as lidars are currently being investigated as alternatives to cup anemometers on meteorological towers for the measurement of wind speed and direction. Although lidars can measure mean wind speeds at heights spanning an entire turbine rotor disk and can be easily moved from one location to another, they measure different values of turbulence than an instrument on a tower. Current methods for improving lidar turbulence estimates include the use of analytical turbulence models and expensive scanning lidars. While these methods provide accurate results in a research setting, they cannot be easily applied to smaller, vertically profiling lidars in locations where high-resolution sonic anemometer data are not available. Thus, there is clearly a need for a turbulence error reduction model that is simpler and more easily applicable to lidars that are used in the wind energy industry.

In this work, a new turbulence error reduction algorithm for lidars is described. The algorithm, L-TERRA, can be applied using only data from a stand-alone vertically profiling lidar and requires minimal training with meteorological tower data. The basis of L-TERRA is a series of physics-based corrections that are applied to the lidar data to mitigate errors from instrument noise, volume averaging, and variance contamination. These corrections are applied in conjunction with a trained machine-learning model to improve turbulence estimates from a vertically profiling WINDCUBE v2 lidar. The lessons learned from creating the L-TERRA model for a WINDCUBE v2 lidar can also be applied to other lidar devices.

L-TERRA was tested on data from two sites in the Southern Plains region of the United States. The physics-based corrections in L-TERRA brought regression line slopes much closer to 1 at both sites and significantly reduced the sensitivity of lidar turbulence errors to atmospheric stability. The accuracy of machine-learning methods in L-TERRA was highly dependent on the input variables and training dataset used, suggesting that machine learning may not be the best technique for reducing lidar TI error. Future work will include the use of a lidar simulator to better understand how different factors affect lidar turbulence error and to determine how these errors can be reduced using information from a stand-alone lidar.

## 1 Introduction

As turbine hub heights increase and wind energy expands to complex and offshore sites, new measurements of the wind resource are needed to inform decisions about site suitability and turbine selection. Currently, most of these measurements

are collected by cup anemometers on meteorological (met) towers. Met towers are fixed in location and typically only collect measurements up to and including the height corresponding to the turbine hub height. However, the measurement of wind speeds across the entire turbine rotor disk is extremely important for power estimation (e.g., Wagner et al., 2009), particularly as modern turbines increase in size. In addition, met towers are expensive to construct and maintain; the estimated cost for

installing and maintaining an 80m land-based met tower for a 2-year campaign is €92,000 (≈ 105,000 USD; Boquet et al., 2010). In response to the limitations of met towers for wind energy, remote sensing devices such as lidars (light detection and ranging) have been proposed as potential alternatives to cup anemometers on towers. Lidars are now frequently used in the research community (e.g., Barthelmie et al., 2013; Stawiarski et al., 2013; Fuertes et al., 2014; Sathe et al., 2015b), and acceptance of lidars in the wind energy community is increasing. The use of remote sensing devices for power performance

testing in flat terrain is discussed in Annex L of the most recent draft version of IEC 61400-12-1 (International Electrotechnical Commission, 2013).

While lidars are capable of measuring mean wind speeds at several different measurement heights (e.g., Sjöholm et al., 2008; Peña et al., 2009; Barthelmie et al., 2013; Sathe et al., 2015b), they measure different values of turbulence than a cup or sonic anemometer (e.g., Sathe et al., 2011; Newman et al., 2016b). Turbulence, a measure of small-scale fluctuations in

the atmospheric flow, is an extremely important parameter in the wind energy industry. Turbulence measurements are used to classify potential wind farm sites and select suitable turbines (International Electrotechnical Commission, 2005) and can also impact power production (e.g., Elliott and Cadogan, 1990; Peinke et al., 2004; Clifton and Wagner, 2014). Because of the paramount importance of turbulence measurements to the wind energy industry, lidars must be able to accurately measure turbulence to be considered a viable alternative to met towers. The inability of lidars to accurately measure turbulence is

currently one of the main barriers to replacing met towers with lidars.

In this work, a new turbulence error reduction model, the Lidar Turbulence Error Reduction Algorithm (L-TERRA), was developed for the WINDCUBE v2 (WC) vertically profiling lidar. The model combines physics-based corrections, such as a spectral correction, with machine-learning techniques to improve estimates of lidar turbulence intensity (TI), defined as the standard deviation of the streamwise wind speed divided by the average wind speed over a 10min period and multiplied by

100%. While the physics-based corrections can be applied using data from the lidar itself, the machine-learning portion of L-TERRA requires training with a collocated lidar/met tower dataset. Unlike other methods for improving lidar turbulence estimates, L-TERRA is a simple method that can be easily applied to vertically profiling lidars. The goal of L-TERRA is to bring lidar TI estimates closer to the values of TI that would be measured by a cup anemometer on a tower. Although cup anemometers are affected by overspeeding (e.g., Kaimal and Finnigan, 1994) and mast distortion (e.g., Wyngaard, 1981), they

provide sufficient information for wind resource assessment and power performance testing and are the current instrument of reference for wind energy.

The manuscript is organized as follows. Section 2 outlines the main factors that affect lidar turbulence estimates and current methods for improving turbulence estimates. A basic description of the different modules in L-TERRA is given in Section 3. The data sets used to train and test L-TERRA are discussed in Section 4. Results from L-TERRA are discussed in Section

5, and a sensitivity analysis is conducted to determine the effects of site conditions on lidar TI error both before and after L-TERRA has been applied. Conclusions and plans for future work are discussed in Section 6.

## 2   Background

Although lidars are frequently used in wind energy studies (e.g., Peña et al., 2009; Krishnamurthy et al., 2013; Clifton et al.,
2015; Wharton et al., 2015; Newsom et al., 2015), they typically measure different values of turbulence than a cup or sonic anemometer (e.g., Sathe et al., 2013; Newman et al., 2016b). In this section, the factors that cause these turbulence discrepancies are discussed. In addition, current methods for reducing turbulence measurement errors from lidars are highlighted. Throughout this work, the process of "correcting" lidar turbulence refers to techniques that are used to bring lidar turbulence estimates closer to the turbulence that would be measured by a cup or sonic anemometer and "error" is used as a synonym for "difference".

### 2.1   Lidar technology

Lidars emit laser light into the atmosphere and measure the Doppler shift of the backscattered energy to estimate the mean wind velocity of volumes of air. Laser light from Doppler lidars is typically scattered by aerosol particles in the atmosphere, which are normally prevalent in the boundary layer (Emeis, 2010). For pulsed Doppler lidars, the time series of the returned signal is split into blocks that correspond to range gates and processed to estimate the average radial wind speed at each range
gate (Huffaker and Hardesty, 1996). In contrast, continuous wave lidars focus the laser beam at different distances from the lidar to estimate wind speeds at different ranges (Slinger and Harris, 2012).

Vertically profiling lidars, which are commonly used in wind energy, involve scanning a cone directly above the lidar to derive the $u$, $v$, and $w$ velocity components. If the atmosphere is horizontally homogeneous in the area enclosed by the cone, the radial velocity measured by the lidar, $v_r$, can be related to the three-dimensional wind components as follows (Weitkamp,
2005):

$$v_r = u\sin\theta\cos\phi + v\cos\theta\cos\phi + w\sin\phi, \tag{1}$$

where $\theta$ is the azimuthal angle of the lidar beam, measured clockwise from north, and $\phi$ is the elevation angle of the lidar beam, measured from the ground. Typically, a raw time series of $u$, $v$, and $w$ is derived from Eq. 1 and these raw wind speed components are used to calculate turbulence parameters. A different method involves taking the variance of Eq. 1 and
combining the radial velocity variance values from multiple beam positions to directly estimate the $u$, $v$, and $w$ variance components (Sathe et al., 2015b).

One lidar that is frequently used in the wind energy industry is the WC, manufactured by Leosphere (Orsay, France). The WC employs a Doppler-Beam swinging (DBS) (e.g., Strauch et al., 1984) technique to estimate the three-dimensional wind vector wherein an optical switch is used to point the laser beam toward the four cardinal directions (north, east, south, and
west) at an angle of 28° from zenith and Eq. 1 is used to derive a time series for $u$, $v$, and $w$. The WC used in this work also

includes a vertical beam position for a direct measurement of the vertical velocity. The WC accumulates measurements at each beam position for just under one second, such that a full scan takes approximately 4–5 seconds. However, velocity data from the WC are updated each time new information is obtained (i.e., every time the beam moves to a different position), leading to an output frequency of 1 Hz.

## 2.2 Errors in lidar data

In Doppler wind lidars, instrument noise results from factors such as the limited amount of aerosol scatterers in the probe volume (Lenschow et al., 2000) and spontaneous radiation emissions from the laser (Chang, 2005). Instrument noise increases the variability of the radial wind speeds measured by the lidar, which artificially increases the turbulence estimates. In contrast, volume averaging decreases the turbulence estimated from the lidar. To obtain a reasonable estimate of the radial velocity, lidars require backscatter data from a large number of scatterers within a probe volume. For the WC, the probe volume measures 20 m along the beam and is negligibly small in the cross-beam and vertical directions. The probe volume acts as a low-pass filter, effectively filtering out all turbulent motions that occur on spatial scales smaller than 20 m. The probe volume is a trade-off between spatial resolution and data accuracy; if the probe volume were smaller than 20 m, fewer data points would be available to estimate the radial velocity, and there would be a higher amount of uncertainty in the measurements.

The scanning strategy used by a lidar can also induce errors in the turbulence estimates. For example, the DBS technique used by the WC requires the assumption that the instantaneous flow field is uniform across the scanning circle. However, this assumption is generally not true in turbulent flow, when the wind field changes significantly in both space and time (e.g., Wainwright et al., 2014; Lundquist et al., 2015). As the WC scanning circle has a diameter of 106 m at a measurement height of 100 m above ground level (AGL), it is likely that the instantaneous flow field changes in space, even in flat terrain. This changing flow field across the lidar scanning circle introduces additional terms into the variance calculations in a phenomenon known as variance contamination (e.g., Sathe et al., 2011; Newman et al., 2016b). This effect contaminates the true value of the velocity variance and can cause the lidar to measure higher values of turbulence than a cup or sonic anemometer.

## 2.3 Current methods for correcting lidar turbulence

Several data processing techniques and state-of-the art measurement configurations have already been developed for acquiring turbulence measurements from lidars (Sathe et al., 2015a). However, many of these measurement configurations require expensive scanning lidars or the fitting of turbulence models that are technically only valid under neutral atmospheric conditions. These techniques are applicable in a research setting, but largely require more instrumentation and measurement data than are typically available during a wind resource assessment.

### 2.3.1 Fitting a turbulence model

One method for correcting lidar turbulence includes modeling the spatial averaging effects of the lidar probe volume. This method involves convolving the true radial velocity field with a spatial weighting function that is controlled by the lidar beam

pattern (e.g., Sjöholm et al., 2009; Sathe et al., 2011). Spatial weighting functions for both pulsed and continuous wave lidars are relatively straightforward (e.g., Sonnenschein and Horrigan, 1971). However, modeling the true velocity field requires knowledge of the three-dimensional turbulence structure, which can be described by the spectral velocity tensor, $\Phi_{ij}$.

The spectral velocity tensor can be modeled through use of the Mann (1994) turbulence model, as in Sjöholm et al. (2009),
5 Mann et al. (2010), Sathe et al. (2011), and others. Fitting the model requires three parameters: a turbulence dissipation rate parameter, a length scale, and a parameter that describes the anisotropy of the flow. Values for these parameters can be estimated by using high-frequency sonic anemometer data and can also be approximated from lidar data. However, the Mann (1994) turbulence model is technically only valid in the surface layer under neutral conditions and is not valid in complex terrain.

### 2.3.2 Six-beam method

To reduce variance contamination caused by the DBS and Velocity-Azimuth Display (VAD; Browning and Wexler, 1968) techniques, Sathe et al. (2015b) proposed a new six-beam scanning technique for Doppler lidars that utilizes the variance of the radial velocity. Newman et al. (2016b) tested the six-beam method with a scanning lidar at the Boulder Atmospheric Observatory in Erie, Colorado, and compared six-beam variance estimates to estimates from sonic anemometers on a tower at the site. Newman et al. (2016b) found that while the six-beam method did generally reduce variance contamination in
comparison to estimates from a WC lidar, errors in the different radial velocity variance estimates caused large errors and even negative values in the resulting $u$ and $v$ variance estimates. Better estimates of the radial velocity variance are likely needed from lidars to obtain accurate results for the six-beam technique.

### 2.3.3 Multiple lidars

While single lidars require measurements around a scanning circle to estimate the three-dimensional velocity field, multiple
scanning lidars can be pointed toward a particular volume of air to obtain turbulence estimates with much higher spatial resolution (e.g., Calhoun et al., 2006; Fuertes et al., 2014; Newsom et al., 2015; Newman et al., 2016a). To collect turbulence measurements, multi-lidar systems must be temporally and spatially synchronized with a high degree of accuracy. Synchronization techniques have been developed for a set of user-customized scanning lidars (Vasiljevic et al., 2014), but are currently not easily implemented on most other scanning lidars. In addition, scanning lidars are much more expensive than commercially
available vertically profiling lidars, particularly if more than one scanning lidar is required for operation.

### 2.3.4 Structure functions

Structure functions describe the spatial correlation of a variable at different separation distances (e.g., Stull, 2000). If the turbulence is isotropic and the turbulence length scale is large, the structure function can be approximated by the Kolmogorov (1941)
model and used to estimate the velocity variance. Krishnamurthy et al. (2011) used scanning lidar data from a field campaign to calculate structure functions in both the along-beam and azimuthal directions and fit the functions to the Kolmogorov (1941)

model to obtain estimates of the velocity variance. The lidar data used by Krishnamurthy et al. (2011) were obtained from a series of plan-position indicator (PPI) scans with high azimuthal resolution, which is typically not available from a scanning strategy used by a vertically profiling lidar.

### 2.3.5 Doppler spectrum

As discussed by Mann et al. (2010), the spectral density of a particular radial velocity, $v_r$, is essentially a weighted count of all the positions within the probe volume where the radial velocity is equal to $v_r$. The weighting occurs because the intensity of the lidar beam is highest at the center of the probe volume and drops off for distances in either direction from the probe volume center. The ensemble-averaged spectrum can then be related to the probability density function of the radial velocity at each position within the probe volume. Given this relation, the unfiltered ("true") variance can be obtained from the second

central moment of the Doppler spectrum. If the lidar is mounted on a turbine nacelle and pointing upstream, as in Branlard et al. (2013), it can be assumed that the wind field is homogeneous along the lidar beam and that the probability density of $v_r$ is approximately uniform along the probe volume. However, if a ground-based, vertically profiling lidar is used, the mean wind field will not be uniform along the lidar's line-of-sight and the effects of shear must be taken into account when estimating the unfiltered variance from the Doppler spectrum (Mann et al., 2010). Currently, this method is more clearly defined for

continuous wave lidars, as the Doppler spectra of pulsed lidars are affected by the finite length of the probe volume in addition to turbulent fluctuations.

### 2.3.6 Summary

Several methods are currently available for obtaining more accurate turbulence estimates from Doppler lidars. Only a few methods were discussed here; a more extensive discussion of turbulence retrieval techniques can be found in Sathe and Mann

(2013) and Sathe et al. (2015a). Most of these methods require the fitting of models and the use of very specific scanning strategies that can currently only be achieved with expensive scanning lidars. The Doppler spectrum method is promising for continuous wave lidars, but requires knowledge of the Doppler spectrum obtained at each lidar beam position, which is usually not available in the output of vertically profiling systems. Thus, there is clearly a need for a turbulence estimation method that can be implemented on vertically profiling lidars that use DBS and VAD techniques and that does not require high-resolution

data from a sonic anemometer. Details of the new turbulence estimation method proposed in this paper are discussed in the next section.

## 3   TI error model: L-TERRA

The TI error model described in this work, L-TERRA, was initially developed for the WC pulsed Doppler lidar. Future work will involve expanding L-TERRA to different lidar configurations and scanning strategies, although the basic framework for

the model will stay the same. The different modules of L-TERRA in its current form are described in this section.

A flowchart depicting different methods for correcting TI with L-TERRA is shown in Fig. 1. Input data to L-TERRA are extracted from the high-frequency output files from the lidar (e.g., the 1 Hz files from the WC). These files typically contain line-of-sight and/or derived $u$, $v$, and $w$ wind speed components that have been estimated from data collected during the lidar's accumulation time at each measurement point. L-TERRA passes these initial data points through several modules that reduce the lidar TI error in different ways. For each of the main modules, outlined in red in Fig. 1, several different methods are available to reduce the TI error, or no method at all can be applied in that module. For example, four different methods were evaluated to reduce noise: a spike filter, and three different methods discussed by Lenschow et al. (2000) (Lenschow 1, Lenschow 2, and Lenschow 3).

Some methods can only be applied to the $u$, $v$, and $w$ velocity data while others can only be applied to radial velocity data, $v_r$; thus, two different model paths can be followed for volume averaging and variance contamination, depending on which wind speed parameters are selected to calculate the variance. In this work, only model combinations that use the $u$, $v$, and $w$ velocity components were evaluated, as not all vertically profiling lidars include the line-of-sight wind speed in the output files. However, methods for both the $u$, $v$, and $w$ components and the radial velocity components are described here for completeness. All possible combinations of the different $u$, $v$, and $w$ methods were tested on each data set to determine which combination produced the largest reduction in TI mean absolute error (MAE).

## 3.1 Preprocessing

Several steps are taken before the lidar data enter the TI correction process. First, values of $u$, $v$, and $w$ are calculated from the raw lidar time series (and values of $v_r$ are extracted from the lidar output files if needed). For the WC lidar, the wind speed components can be calculated in two different ways: by estimating new $u$, $v$, and $w$ components every time the lidar beam moves to a new position (i.e., just under 1 s) or by estimating a single value of each of the wind components for every 4-s scan, similar to a VAD technique. In Section 5, both the 1-s and 4-s techniques are used to calculate the wind components in the evaluation of L-TERRA.

Next, the data are interpolated to a grid with constant temporal spacing (e.g., 1 Hz for the 1-s scans and 0.25 Hz for the 4-s scans), as statistical measures such as the calculation of variance and spectra require that the frequency resolution of the measurements is constant. The mean horizontal wind speed and shear parameter are calculated before L-TERRA is applied, as these parameters are required for implementation of L-TERRA and are relatively unaffected by the errors that plague turbulence measurements.

The 10min mean horizontal wind speed, $\overline{U}$, is defined by Eq. 2:

$$\overline{U} = \overline{(u^2 + v^2)^{1/2}}, \tag{2}$$

where $u$ and $v$ are the east-west and north-south wind components, respectively, and the overbar denotes temporal averaging. The shear parameter, $\alpha$, is derived from the standard power law equation (International Electrotechnical Commission, 2005):

$$U(z) = U(z_r) \left( \frac{z}{z_r} \right)^\alpha,$$ (3)

where $z$ is height above ground and $z_r$ is a reference height. Eq. 3 can be simplified by letting $U(z_r)z_r^{-\alpha}$ equal a constant $\beta$, as in Clifton et al. (2013). The power law then becomes the following:

$$U(z) = \beta z^\alpha$$ (4)

A 10min mean value of $\alpha$ can be found by taking the natural logarithm of Eq. 4 and fitting the resulting equation to a straight line. In this work, values of $\overline{U}$ measured by the WC between 40 and 200 m were used to calculate values of $\alpha$.

The raw wind speeds are rotated into a new coordinate system by forcing $\overline{v}$ and $\overline{w}$ to 0 and aligning $u$ with the 10min mean wind direction (e.g., Kaimal and Finnigan, 1994). The TI is then defined by Eq. 5:

$$TI = \left( \frac{\sigma_u}{\overline{u}} \right) \times 100\%,$$ (5)

where $\sigma_u$ is the standard deviation of $u$ over a 10min period, defined in the new coordinate system, and $\overline{u}$ is the 10min mean wind speed. Eq. 5 gives the initial lidar-estimated value of the horizontal TI. The same procedure was used to calculate TI from the sonic anemometer data used in this work. Cup anemometer TI was calculated using the mean horizontal wind speed and standard deviation in the cup anemometer output data stream. As the main purpose of this work is to bring lidar TI estimates closer to point measurements from any type of reference device on a met tower, the difference in the way TI is calculated for cup and sonic anemometers is not of paramount importance.

## 3.2 Physics-based corrections

The next three modules comprise the physics-based corrections in L-TERRA. These corrections rely only on data from the lidar itself and use meteorological theories to apply corrections to the TI estimates.

### 3.2.1 Instrument noise

After the lidar data are processed, different techniques are used to remove noise and spurious data. A standard way to remove outliers from a time series is to use a spike filter (e.g., Vickers and Mahrt, 1997). A basic spike filter was evaluated for the model in addition to several methods developed by Lenschow et al. (2000).

The spike filter routine used in this work was based on one of the lidar pre-processing steps presented by Wang et al. (2015). First, the difference between adjacent velocity values, $\Delta v$, is calculated for all velocity measurements in a 10min period, defined by the following equation:

$$\Delta v_i = v_{i+1} - v_i. \tag{6}$$

A data point $v_i$ is defined as a spike and removed from the dataset if the following conditions are met:

$$
\begin{aligned}
|\Delta v_i| &> 2IQR_{\Delta v}, \\
|\Delta v_{i-1}| &> 2IQR_{\Delta v}, \\
\frac{\Delta v_i}{\Delta v_{i-1}} &= -1
\end{aligned}
\tag{7}
$$

where $IQR_{\Delta v}$ is the interquartile range of all values of $\Delta v$ within the 10min period. This spike filter eliminates data points defined by a large decrease in velocity followed by a large increase in velocity (or vice versa).

The Lenschow et al. (2000) methods involve the use of the lidar's velocity spectrum or autocovariance function to estimate the amount of noise in the variance measurements from the lidar. In the lidar velocity spectrum, power at the high frequency end of the spectrum is assumed to be largely attributed to white noise. The average power at the high frequency end of the spectrum is integrated across all frequencies to estimate the variance due to noise. An estimate of noise variance can also be made by assuming that noise is random and completely uncorrelated with the velocity signal. Thus, the value of the autocovariance function at lag 0 is a sum of the noise variance and actual velocity variance. The noise variance can be estimated by extrapolating the autocovariance function from non-zero lags to lag 0. The difference between the autocovariance function at lag 0 and the extrapolated value is the noise variance.

### 3.2.2 Volume averaging

Two methods were considered to mitigate the effects of volume averaging: structure functions and spectral extrapolation. As discussed in Section 2.3.4, structure functions can be estimated using available lidar data and fit to models to estimate turbulence parameters (e.g., Krishnamurthy et al., 2011). By fitting the lidar data to a model, the reduction of turbulence due to volume averaging is mitigated. Although the estimation of structure functions with a lidar is optimized with the use of a high-resolution PPI scan, as in Krishnamurthy et al. (2011), structure functions can also be estimated from DBS scans. A longitudinal (i.e., along-beam) structure function can be estimated from a DBS scan by using velocity data from different range gates along the same radial. An azimuthal structure function can be estimated by combining data from different azimuthal directions at the same height, although the fit is likely to be poor for a DBS scan due to the small number of azimuthal angles where data are collected.

Another method of mitigating volume averaging is to model the lidar velocity spectrum and use the model to extrapolate the spectrum to higher frequencies. The high-frequency part of the modeled spectrum can then be integrated to obtain an estimate

of the variance that is not measured by the lidar as a result of spatial and temporal resolution (e.g., Hogan et al., 2009). The standard Kaimal spectrum used for wind energy (e.g., Burton et al., 2001) requires three parameters for fitting: the mean wind speed, the variance, and a length scale. The first two parameters can be calculated from the lidar data directly, while the last parameter must be estimated. In this work, the length scale was estimated in two different ways: by minimizing the difference between the actual velocity spectrum and the Kaimal spectrum (Spectral Fit 1) and by calculating the integral time scale from the raw velocity time series, which can then be related to the integral length scale through Taylor's frozen turbulence hypothesis (Spectral Fit 2).

### 3.2.3 Variance contamination

Methods to reduce variance contamination include the six-beam technique developed by Sathe et al. (2015b), discussed in Section 2.3.2, and the use of Taylor's frozen turbulence hypothesis with data from the WC's vertical beam (Newman, 2015). The six-beam technique can be adapted for the WC DBS scans by estimating the variance from the five different radial beam positions (four off-vertical and one vertical) and solving a system of five equations to determine the variance and co-variance components. Although the covariance of the $u$ and $v$ components cannot be determined with this method, the three velocity variance components can be estimated (Newman et al., 2016b). Taylor's frozen turbulence hypothesis can be used to relate temporal changes in the vertical velocity measured by the WC's vertical beam to spatial changes in the vertical wind field across the WC scanning circle. Spatial changes in the $w$ component can then be used to reduce contamination in either the raw wind speed (Taylor 1) or the variance directly (Taylor 2).

### 3.3 Machine learning

The last module in L-TERRA uses a trained machine learning model to further reduce TI error. Inputs for the machine learning module include lidar-measured parameters (e.g., mean wind speed and shear) and the corrected TI produced by the physics-based corrections. The model must first be trained on one or more datasets that contain data from a collocated met tower and lidar.

Two machine-learning methods were evaluated as part of L-TERRA: random forests and multivariate adaptive regression splines (MARS). The use of other machine learning techniques such as neural networks could be an area of future research, though we decided to focus mainly on the physical corrections for this work. Random forest models are developed by averaging multiple decision trees that were trained on different subsets of the data. By averaging tens or hundreds of decision trees, the variance of the overall model is reduced significantly (Friedman et al., 2001). Random forests were evaluated because they are relatively easy to understand and have previously been used for wind energy applications (e.g., Clifton et al., 2013; Bulaevskaya et al., 2015). MARS is essentially a stepwise regression model, where different coefficients and basis functions are used to predict the output depending on each region in the dataset (Friedman, 1991). MARS is well-suited for the prediction of physical processes due to its ability to model non-linearities and interactions among variables.

Potential predictor variables for the machine-learning models were divided into two broad categories: atmospheric state and lidar operating characteristics. Variables that were evaluated as predictor variables in L-TERRA are given in Table 1.

Atmospheric state variables included shear parameter, mean wind speed, Doppler spectral broadening, and $u$ and $w$ velocity variances. Lidar operating characteristics included signal-to-noise ratio (SNR) and internal instrument temperature. In all, 18 predictor variables were considered for the machine learning portion of L-TERRA.

### 3.4 Comparison to previous methods

In contrast to the methods discussed in Sect. 2.3, L-TERRA uses only information that is available from a standard vertically profiling lidar. The physics-based corrections in L-TERRA require only data from the lidar itself, while the machine learning module in L-TERRA can be trained using either cup or sonic anemometer data. The majority of the corrections in L-TERRA can be implemented with fewer than 20 lines of code, and the models employed in L-TERRA are well-documented in the literature and simple to understand. It takes approximately 0.1 seconds to run L-TERRA for a single 10min period, making it easy to implement in real time. As discussed in Sect. 5.1.2, a stability-dependent version of L-TERRA can be used to adapt to changing conditions and apply corrections appropriate for the current atmospheric stability regime.

## 4   Data sets

### 4.1   Measurement sites

L-TERRA was tested on data from two different locations: the Southern Great Plains ARM site in Lamont, Oklahoma (Fig. 2), and an operational wind farm in the Southern Plains region of the United States. At both sites, a WC lidar was deployed for field campaigns lasting several months and was configured to collect measurements at heights corresponding to reference instruments.

The ARM site, a field measurement site operated by the U.S. Department of Energy, contains several remote sensing and in-situ instruments (Mather and Voyles, 2013). From November 2012 to June 2013, a WC lidar owned by Lawrence Livermore National Laboratory was deployed at the ARM site approximately 100 m from a 60m tower. Gill Windmaster Pro 3-D sonic anemometers are mounted on the tower at 25 and 60 m AGL and collect velocity data at a frequency of 10 Hz. The ARM site is relatively flat, with maximum elevation changes of approximately 5 m in the surrounding area. The land to the east of the tower slopes gently upward toward the ARM site (Fig. 2), although few data points from this sector were used to evaluate L-TERRA, as the sonic anemometers are located on the west side of the tower and are thus affected by tower wakes when winds are from this direction.

The WC was also deployed at an operational wind farm in the Southern Great Plains. (Due to a nondisclosure agreement with the wind farm, we cannot disclose the exact location of the wind farm.) Similar to the ARM site, the wind farm is located in fairly flat terrain with maximum elevation differences of 5–10 m in the regions surrounding the WC deployments. The WC was located on the wind farm from November 2013 to July 2014, with a break from February to April 2014 while the WC was deployed for a different field experiment. During the wind farm deployments, the WC was sited in the same enclosure as a met tower with standard wind energy instrumentation, including a cup anemometer at 80 m. For the winter deployment, the WC

was located near a met tower on the north end of the wind farm, and for the spring/summer deployment, the WC was moved to a tower enclosure at the south end of the wind farm, in accordance with the dominant wind direction during each season at the wind farm. Directional sectors that may have been influenced by nearby turbines were determined following the guidelines in Annex A of IEC 61400-12-1 (International Electrotechnical Commission, 2013) and were excluded from the dataset.

## 4.2 Stability classification

Typical atmospheric stability parameters include the gradient Richardson number ($Ri$) and the Monin-Obukhov length ($L$) (e.g., Stull, 2000). Calculation of $Ri$ requires temperature and wind speed measurements at two different heights while calculation of $L$ requires high-frequency flux measurements at a single height. As the goal of L-TERRA is to apply TI corrections to a standalone lidar, we preferred to classify stability depending only on measurements available from a lidar.

One option for a lidar-based stability parameter is the shear exponent, $\alpha$ (Eq. 4). Although $\alpha$ can change with height or surface roughness (e.g., Petersen et al., 1998), it is strongly tied to the atmospheric stability in the Central and Southern Plains regions of the United States (e.g., Walter et al., 2009; Vanderwende and Lundquist, 2012; Newman and Klein, 2014). This relation is likely apparent because the diurnal transition of the atmospheric boundary layer largely controls the wind speed profile in flat terrain (e.g., Arya, 2001).

To test the ability of the lidar shear exponent to classify stability, values of $\alpha$ calculated with WC data between 40 and 200 m were compared to values of $Ri$ calculated from 4 and 60m wind speed and temperature data from the ARM site. $Ri$ was estimated with the following equation (Bodine et al., 2009; Newman and Klein, 2014):

$$Ri = \frac{g[(T_{60m} - T_{4m})/\Delta z_T + \Gamma_d]\Delta z_U^2}{T_{4m}(U_{60m} - U_{4m})^2} \qquad (8)$$

where $g$ (m s$^{-2}$) denotes the gravitational acceleration, $T$ (K) is the temperature, $U$ (m s$^{-1}$) is the mean horizontal wind speed, $\Gamma_d$ (K m$^{-1}$) is the dry adiabatic lapse rate, and $\Delta z_T$ and $\Delta z_U$ (m) represent the difference in measurement height for values of $T$ and $U$, respectively. Thresholds for $\alpha$ are loosely based on the thresholds used in Wharton and Lundquist (2012) and are given in Table 2. Thresholds for $Ri$ were -0.17 for the transition between unstable and neutral conditions and 0.06 for the transition between neutral and stable conditions, as in Vanderwende and Lundquist (2012).

A scatter plot of $Ri$ versus $\alpha$ for the ARM site is shown in Fig. 3. (Only 30min temperature data were available from the tower, so 30min, rather than 10min, values of $Ri$ and $\alpha$ are shown.) Of the 558 30min time periods represented in Fig. 3, the same stability classifications were made based on both $Ri$ and $\alpha$ approximately 52.5% of the time. However, many of the incorrect classification periods occurred during near-neutral conditions, with $Ri$ values near 0 and/or $\alpha$ values near the neutral standard of 1/7 ($\approx$0.143). Opposite classifications were made in 5% of the cases shown in Fig. 3. The potential impact of these stability misclassifications on TI correction methods is discussed in Section 5.1.3.

### 4.3 Comparison of mean wind speed and TI

At both sites, 10min mean wind speeds measured by the WC and the met tower instruments were well-correlated, with regression line slopes of approximately 1 and $R^2$ values of approximately 0.99 (not shown). Thus, we felt confident that the WC was measuring similar conditions to the reference instruments, though a modified version of L-TERRA could be used in the future to mitigate any small errors in measurement of mean wind speeds. Larger discrepancies were evident in the comparison of TI values, which is our current area of focus for L-TERRA. Sample scatter plots of met tower versus lidar TI for the ARM site are shown in Fig. 4a, and corresponding regression statistics for the raw TI are shown in Table 3.

At the ARM site, $\alpha$ was strongly related to the sign of TI errors, with the WC overestimating TI under unstable conditions and underestimating TI under stable conditions (Fig. 4a). The over- and underestimation of TI was likely due to the effects of variance contamination and volume averaging, respectively. Regression line slopes increase with decreasing stability (Table 3), which is consistent with previous modeling and observational studies in flat terrain (e.g., Sathe et al., 2011; Rodrigo et al., 2013; Schneemann et al., 2014). Initial TI error trends from the wind farm data set are quite similar to those found in the ARM data set (Table 4). As TI error trends based on $\alpha$ were consistent with previous work that classified lidar variance errors by stability, we felt confident in using $\alpha$ as a proxy for stability for the datasets in this work.

## 5   L-TERRA results

First, data from each site were examined individually to assess the performance of L-TERRA. Results from the physics-based corrections were analyzed separately from results from the full version of L-TERRA (physics-based corrections plus machine learning) to assess how well each set of corrections performed.

### 5.1 Application of physics-based corrections

#### 5.1.1 Initial version of L-TERRA

For both the ARM site and the wind farm, all possible combinations of the physics-based corrections described in Section 3.2 for the $u$, $v$, and $w$ components were evaluated. Initially, the model combination that minimized the overall TI MAE was selected as the optimal model combination for that particular site. Data were filtered to avoid mast shadowing, and 10min periods where the mean wind speed was less than 4 m s$^{-1}$ were not used to evaluate L-TERRA, as the standards outlined in IEC 61400-12-1 (International Electrotechnical Commission, 2013) restrict remote sensing classification to wind speeds between 4 and 16 m s$^{-1}$.

At the ARM site, the original TI MAE was 1.5%, and MAE values that resulted from application of L-TERRA ranged from 1.31% to 2.73%. MAE values above the original value of 1.5% indicate that for these model combinations, L-TERRA actually increased overall error in WC TI. For many of these model combinations with high MAE values, the MAE increased for stable conditions while decreasing for unstable conditions, and vice versa, indicating that some model combinations work better for particular stability conditions than others. Many of the high MAE values were also associated with model combinations that

used the Lenschow noise removal techniques with the 4-s VAD scans. The Lenschow techniques are more aggressive with noise removal than a spike filter and also involve directly reducing the variance due to noise, rather than removing spikes in the raw velocity time series. It is possible that the noise apparent in the original WC data artificially brought the WC TI estimates closer to the sonic TI, and removing the noise variance decreased the WC TI values, bringing them further from the sonic values and increasing the MAE. In addition, spectra and autocovariance functions derived from the 4-s data are expected to be less accurate than those derived from the higher-resolution 1-s data, which could affect the accuracy of the Lenschow techniques.

TI MAE values also varied a large amount at the wind farm, with an original MAE of 1.46% and L-TERRA MAE values ranging from 1.38% to 2.9%. Similar to the ARM site, many of the higher MAE values were associated with model combinations that decreased the MAE for certain stability conditions while increasing the MAE for other stability conditions.

Overall, the model combination that minimized the TI MAE was nearly the same for both sites and is shown in the first row of Table 5 (the only difference between the optimal model combination at the sites was in the variance contamination module). Application of this initial L-TERRA model combination resulted in a modest reduction of lidar TI MAE from 1.5% to 1.31% at the ARM site and from 1.46% to 1.38% at the wind farm (not shown). Several slightly different model combinations produced similar MAE values at both sites, suggesting that there may actually be multiple optimal combinations of L-TERRA at each site when the MAE is minimized. For example, at the wind farm, the minimum MAE of 1.38% was achieved when the first Lenschow method was used to reduce noise from the raw 1 Hz wind speeds and the first correction using Taylor's frozen turbulence hypothesis was employed to reduce variance contamination. However, MAE values within 0.01% of 1.38% were also achieved when different noise removal techniques were used instead of the first Lenschow technique, or when no noise removal technique was used at all. Thus, at this site, noise removal appeared to have a very minor impact on the WC TI values. It may be useful to consider other parameters in determining the optimal model combination, such as regression line statistics or sensitivity of TI error to atmospheric stability. However, minimizing the MAE is a standard approach for determining optimal model combinations and provides a useful baseline combination for evaluating L-TERRA.

### 5.1.2 Stability-dependent version of L-TERRA

By examining the change in lidar TI after each step in L-TERRA, it was determined that some corrections decreased error under stable conditions but increased error under unstable conditions, and vice versa. This is not surprising, as the magnitude and sign of TI errors was strongly dependent on atmospheric stability at both sites (Tables 3, 4) as a result of the different factors that affect TI error under different stability conditions. Thus, optimal model combinations were next determined separately for the three different bulk stability classes to form a stability-dependent version of L-TERRA (L-TERRA-S). Optimal model combinations were very similar for both sites and are shown in Table 5.

For stable and unstable conditions, a spike filter was the optimal noise removal technique. Only the model chain for stable conditions included a volume averaging correction, likely because volume averaging effects on TI are largest under stable conditions. For unstable conditions, using the velocity time series from the VAD technique produced the largest reduction in MAE. While the raw output time series from the WC is available at 1 Hz (Section 2.1), the VAD technique is typically applied

once per full scan to derive the three-dimensional wind vector. For the WC, this results in an output data frequency of 0.25 Hz for the VAD technique. The lower temporal resolution of the VAD technique likely served to artificially reduce some of the effects of variance contamination, as smaller scales of turbulence were not measured.

The impact of each of the different physics-based correction modules on the TI error is shown in Fig. 5. Overall, MAE steadily decreased after application of the different corrections, with the largest decrease occurring for the volume averaging module. For unstable TI values, the noise reduction module had the largest impact on reducing MAE and bringing the regression line slope closer to 1. The variance contamination module served to further reduce the MAE and regression line slope, bringing the slope from approximately 1.05 to 1.00, but the $R^2$ value of the regression line decreased. Similarly, the variance contamination module reduced the regression line slope for neutral TI values, bringing it closer to 1, but caused the $R^2$ value of the regression line to decrease. This resulted in an MAE value of 1.48% for neutral TI values after L-TERRA was applied, which is the same as the original MAE value for neutral conditions. The corrections performed best on stable TI values, with MAE values steadily decreasing and $R^2$ values increasing with each correction. The volume averaging module caused the largest change in stable TI values, with the regression line slope increasing from 0.90 to 1.01 after application of the volume averaging module. In summary, all the physics-based corrections in L-TERRA appear necessary to the correction of WC TI, though the importance of each correction depends on the stability. The variance contamination module likely needs to be improved for certain types of unstable and neutral conditions, as it increased TI error for some unstable and neutral TI values.

Scatter plots of ARM site TI data after L-TERRA-S was applied are shown in Fig. 4b and corresponding regression statistics are shown in Table 3. L-TERRA-S served to bring the majority of WC TI estimates closer to the one-to-one line, resulting in regression line slopes of 1.00, 1.01, and 1.00 for stable, neutral, and unstable conditions, respectively. In addition, the overall TI MAE decreased from 1.5% to 1.25%. However, as discussed in the previous paragraph, $R^2$ values for neutral and unstable conditions decreased slightly. Thus, although L-TERRA-S improved the accuracy of most lidar TI estimates, it also increased scatter for neutral and unstable conditions.

Results for the wind farm were similar, with overall MAE decreasing from 1.46% to 1.19% and regression line slopes becoming 1.00, 1.04, and 1.05 for stable, neutral, and unstable conditions, respectively (Table 4). $R^2$ values for neutral and unstable conditions also decreased slightly.

### 5.1.3 Effects of stability misclassification

One possible explanation for the increase in scatter at both sites could be misclassification of atmospheric stability by the shear exponent, $\alpha$. As discussed in Sect. 4.2, opposing stability classifications were made based on $Ri$ and $\alpha$ during approximately 5% of the 30min cases where both stability parameters were available. For L-TERRA-S, an incorrect stability classification could lead to application of corrections that are not well-suited for the actual errors in the lidar TI. To examine the impact of incorrect stability classification on remaining TI errors, time periods were identified where the WC TI error at the ARM site was above the 95th percentile of all TI errors after L-TERRA-S was applied; these TI values represent outliers in Fig. 4b and contribute significantly to the low values of $R^2$.

Of the 56 TI outliers identified that were associated with valid values of $Ri$ and $\alpha$, 16 points were classified as unstable by both $Ri$ and $\alpha$, 5 points were classified as neutral by both parameters, and 4 points were classified as stable by both parameters. The remaining 33 points were classified differently by $Ri$ and $\alpha$, with nearly half (14) of the points being classified as neutral by $Ri$ and slightly unstable or slightly stable by $\alpha$, 12 points being classified as stable or unstable by $Ri$ and neutral by $\alpha$, and the remaining 5 points being classified as stable by $Ri$ and slightly unstable by $\alpha$. The majority of these stability misclassifications appear to occur near neutral conditions, where small errors in measurement of the wind shear or temperature gradient could lead to a different stability classification. Thus, it may be useful in the future to use a blend of correction techniques for points classified as near-neutral or use additional parameters to classify stability from lidar data. However, as opposing stability classifications accounted for fewer than 10% of the large TI errors apparent after application of L-TERRA-S, it is likely that other factors were also responsible for the large amount of scatter still apparent in the TI data. For example, it is likely that the current physics-based corrections in L-TERRA-S do not fully capture all the factors that affect lidar TI error, resulting in large errors that remain for some data points. In the next section, machine-learning techniques are evaluated as a potential method to model the remaining physics that impact lidar TI error.

## 5.2 Application of machine-learning techniques

The physics-based corrections described in the previous section require only data from the lidar itself and do not require any training data. In contrast, machine-learning models must be trained on a dataset before being applied to new data. Typically, a single dataset is split into training and testing datasets in a method known as cross-validation (e.g., Efron and Gong, 1983) to test the accuracy of the model on data that was not included in the training process. As the end goal of L-TERRA is to provide accurate lidar TI values at a site that doesn't have a met tower, the machine-learning model in L-TERRA must be trained on one or more sites with a met tower before being applied to a lidar at a new site. Thus, the machine-learning models discussed in this section were trained on the wind farm data and then applied to data from the ARM site for validation.

### 5.2.1 Determination of predictor variables

To determine appropriate predictor variables for the machine-learning module, a sensitivity analysis was conducted for the WC TI error at both sites. Sensitivity of the lidar TI error to the various predictor variables in Table 1 was quantified following the guidelines in Annex L of the new committee draft of IEC 61400-12-1 (International Electrotechnical Commission, 2013). First, predictor variables were binned and bin-means of the TI percent error corresponding to each bin were calculated. A least-squares technique was then used to calculate a regression line between the predictor bin centers and the bin-means of the TI percent error. Sensitivity, defined as the product of the regression line slope and the standard deviation of the predictor variable, was then calculated for each predictor. The sensitivity gives the approximate change in the TI error for a change in the predictor variable that is equivalent to one standard deviation of the variable.

Sample plots showing the response of TI percent error to different variables at the ARM site are shown in Fig. 6. Raw WC TI error was extremely sensitive to the four variables depicted in Fig. 6, with larger TI percent errors for lower wind speeds and SNR values, and TI errors changing sign from negative to positive for decreasing shear parameter values and increasing raw

TI values. After L-TERRA-S was applied, TI error sensitivity to shear parameter and raw lidar TI decreased significantly. This decrease in sensitivity demonstrates a major advantage of L-TERRA-S, as it implies that lidar TI error is no longer strongly dependent on atmospheric stability. However, while L-TERRA-S decreased sensitivity slightly for mean wind speed and SNR, a high dependence of TI error on these two variables is still apparent in Figs. 6c and 6d.

Overall, the six variables for the ARM site with the highest sensitivity values after application of L-TERRA-S were as follows: integral time scale (vertical), SNR, corrected TI, integral time scale (horizontal), mean wind speed, and shear parameter. (For highly correlated variables, the variable with a higher sensitivity was retained in the list.) These six variables were then used to train a random forest with the wind farm data.

### 5.2.2 Results from trained machine learning model

Results from application of the trained random forest on the ARM site L-TERRA-S TI values are shown in Fig. 7a. Application of the random forest increased MAE values from 0.78% to 0.89% for stable conditions and from 1.48% to 1.6% for neutral conditions, and decreased the MAE from 1.59% to 1.53% for unstable conditions in comparison to the results from L-TERRA-S. For all three stability classifications, $R^2$ values dropped significantly and a positive bias was induced for low TI values.

To determine the cause of this positive bias, the sensitivity values from both sites were compared for the six input variables. While the regression line slope and sensitivity values for the vertical integral time scale, SNR, mean wind speed, and shear were very similar at both sites, sensitivity values for the horizontal integral time scale and corrected TI differed substantially. In particular, the sensitivity of TI error to the corrected TI was 6.86% at the ARM site and over five times larger at 39.6% at the wind farm. After removal of the horizontal integral time scale and the corrected TI from the input parameter list, the bias at low TI values largely disappeared (Fig. 7b), suggesting that the positive bias was related to the different sensitivities associated with two of the input parameters. However, even with these two parameters removed from the input parameter list, the MAE values still increased in comparison to L-TERRA-S while $R^2$ values decreased. Results from the MARS model were similar to those from the random forest models.

These results highlight two major limitations of using machine-learning techniques to improve lidar TI accuracy in L-TERRA: 1) The most significant input parameters can change from one site to another and will not be known a priori for a new site and 2) Sensitivity of TI error to different input variables depends on the training site and the particular lidar and reference measurements used. To investigate the effect of the training dataset used, a random forest was also trained on 75% of the ARM site data and then applied to the remaining 25%. Training and testing a random forest on data from the same site did decrease MAE values in comparison to results from L-TERRA-S, but $R^2$ values still decreased slightly for neutral and unstable conditions. More importantly, using this technique would preclude L-TERRA from being applied at a new site that doesn't have a met tower.

Although machine learning can be a useful tool for turbine power prediction (e.g., Clifton et al., 2013), it does not appear to be an ideal technique for correcting lidar TI error. Thus, next steps in the development of L-TERRA will involve further refining the physics-based corrections in L-TERRA-S to improve TI estimates in a more robust manner. Rather than relying on modeled patterns, physics-based corrections directly relate lidar measurements to TI errors and substantially improved the

accuracy of lidar TI estimates at both sites evaluated in this work (Fig. 4, Tables 3 and 4). However, the current physics-based corrections do not completely eliminate TI error, indicating that the physics that cause TI error are not being entirely captured in L-TERRA-S. Future work will involve the development of a lidar uncertainty framework that outlines all possible causes of lidar error. Different parts of the framework could then be quantified through the use of a simulated flow field and virtual lidar, as in Lundquist et al. (2015).

Results from the sensitivity analysis conducted in this section will greatly assist in determining areas of focus for the lidar uncertainty framework. For example, TI error at both sites was extremely sensitive to the integral time scale of the $w$ wind component, which is a proxy for the dominant temporal scale of turbulent eddies in the vertical direction. Thus, lidar TI error appears to be strongly affected by the scales of vertical motion present in the area enclosed by the lidar scanning circle, which will contribute to the degree of variance contamination. Currently, no physical models exist that could account for these effects, and so we suggest that this could be a fruitful research direction. In future work, the virtual lidar tool will be used to examine how changes in the vertical flow field across the lidar scanning circle impact TI estimates and how information from the lidar can be used to approximate and remove these effects.

## 6   Conclusions and future work

Lidars are currently being considered as a replacement to meteorological towers in the wind energy industry. Unlike met towers, lidars can be easily deployed at different locations and are capable of collecting wind speed measurements at heights spanning the entire turbine rotor disk. However, lidars measure different values of TI than a cup or sonic anemometer, and this uncertainty in lidar TI estimates is a major barrier to the adoption of lidars for wind resource assessment and power performance testing. In this work, a lidar turbulence error reduction model, L-TERRA, was developed and tested on WC lidar data from two different sites. The model incorporates both physics-based corrections and machine-learning techniques to improve lidar TI estimates.

The main findings from this work can be summarized as follows:

- The difference between TI measured by a cup or sonic anemometer and that measured by a vertically-profiling lidar can be reduced using appropriate physical models of the lidar measurements.

- Performance of L-TERRA improves substantially when different model configurations are used for different stability conditions (i.e., in L-TERRA-S).

- In addition to reducing MAE, L-TERRA-S also reduces sensitivity of lidar TI error to atmospheric stability.

- The accuracy of a machine-learning method in L-TERRA-S is highly dependent on the sensitivity of the lidar TI error to the input parameters at both the training and testing sites.

Further improvements to L-TERRA-S can be made through a better understanding of how atmospheric conditions and lidar operating characteristics impact TI error. This understanding can be achieved through development of a lidar uncertainty framework and testing of the framework with modeled atmospheric data. Future work on L-TERRA-S will also include testing

with additional datasets, including data sets from complex terrain and different areas of the world. Practical applications of L-TERRA for site assessment and power prediction will also be explored in future work.

The development of L-TERRA and other TI correction techniques has significant implications for the wind energy industry, which has traditionally relied on data from fixed met towers. L-TERRA can be applied to vertically profiling lidars that are commonly used in the wind industry, thus expanding the use of lidars for wind energy applications. Lidars with improved TI estimates can be used for wake characterization, site classification, power curve testing, site monitoring, and resource assessment. Improved lidar TI estimates could also help wind energy developers make more informed decisions about turbine selection and wind farm layout. The use of lidars in place of met towers for wind energy applications should allow for more rapid development of wind in regions where it is difficult or costly to install met towers, and the improvement of lidar turbulence estimates will greatly assist in the adoption of lidars in the wind industry.

*Acknowledgements.* The authors would like to thank the staff of the Southern Great Plains ARM site and the Southern Plains wind farm for assisting with the lidar field deployments. Sebastien Biraud and Marc Fischer of Lawrence Berkeley National Laboratory supplied sonic anemometer data for the ARM site. Sonia Wharton of Lawrence Livermore National Laboratory provided the WINDCUBE lidar used in this work and assisted with field deployments. Leosphere and NRG Systems provided technical support for the WINDCUBE lidar during the experiments. Conversations with Caleb Phillips at NREL greatly enhanced our understanding of different machine learning models. We also thank Rozenn Wagner and an anonymous reviewer whose comments helped improve the manuscript. The ARM Climate Research Facility is a U.S. Department of Energy Office of Science user facility sponsored by the Office of Biological and Environmental Research. This work was supported by the U.S. Department of Energy under Contract No. DE-AC36-08GO28308 with the National Renewable Energy Laboratory. Funding for the work was provided by the DOE Office of Energy Efficiency and Renewable Energy, Wind and Water Power Technologies Office.

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

| Potential Predictor Variables | |
|---|---|
| **Atmospheric State** | **Lidar Operating Characteristics** |
| Original TI | SNR |
| Corrected TI | Instrument pitch |
| $\sigma_u^2$ | Instrument roll |
| $\sigma_w^2$ | Instrument internal temperature |
| $\overline{U}$ | |
| $\alpha$ | |
| Horizontal wind speed dispersion | |
| Vertical wind speed dispersion | |
| Spectral broadening | |
| Integral time scale (horizontal) | |
| Integral time scale (vertical) | |
| Stationarity (e.g., Vickers and Mahrt, 1997) | |
| Maximum instantaneous value of $w$ | |
| Precipitation | |

**Table 1.** Potential predictor variables evaluated in the machine-learning module of L-TERRA.

| **Shear Parameter Range** | **Stability Classification** |
|---|---|
| $\alpha \geq 0.2$ | Stable |
| $0.1 \leq \alpha < 0.2$ | Neutral |
| $\alpha < 0.1$ | Unstable |

**Table 2.** Stability classifications used in this work.

| | *MAE* | | *Slope* | | $R^2$ | |
|---|---|---|---|---|---|---|
| | **Raw** | **L-TERRA-S** | **Raw** | **L-TERRA-S** | **Raw** | **L-TERRA-S** |
| **Stable (*N* = 1246)** | 1.02 | 0.78 | 0.90 | 1.00 | 0.88 | 0.89 |
| **Neutral (*N* = 590)** | 1.48 | 1.48 | 1.04 | 1.01 | 0.89 | 0.87 |
| **Unstable (*N* = 1322)** | 1.95 | 1.59 | 1.12 | 1.00 | 0.79 | 0.77 |
| **All (*N* = 3158)** | 1.50 | 1.25 | 1.05 | 1.00 | 0.86 | 0.86 |

**Table 3.** Mean absolute error (MAE) and slope and $R^2$ values of regression lines for WC TI compared to met tower TI before and after L-TERRA-S has been applied for the 60m measurement height at the ARM site.

|  | MAE | | Slope | | $R^2$ | |
|---|---|---|---|---|---|---|
|  | Raw | L-TERRA-S | Raw | L-TERRA-S | Raw | L-TERRA-S |
| **Stable ($N$ = 1866)** | 1.13 | 0.88 | 0.90 | 1.00 | 0.92 | 0.92 |
| **Neutral ($N$ = 856)** | 1.22 | 1.14 | 1.07 | 1.04 | 0.87 | 0.85 |
| **Unstable ($N$ = 1771)** | 1.94 | 1.56 | 1.14 | 1.05 | 0.84 | 0.78 |
| **All ($N$ = 4493)** | 1.46 | 1.19 | 1.07 | 1.03 | 0.88 | 0.87 |

**Table 4.** As in Table 3, but for Southern Plains wind farm data.

| Stability Classification | Wind Speed Frequency | Noise | Volume Averaging | Variance Contamination |
|---|---|---|---|---|
| All | 1 Hz | Lenschow 1 | None | Taylor 2 |
| Stable | 1 Hz | Spike filter | Spectral Fit 2 | Taylor 1 |
| Neutral | 1 Hz | None | None | Taylor 2 |
| Unstable | 0.25 Hz | Spike filter | None | Taylor 1 |

**Table 5.** L-TERRA model combinations that minimized TI MAE for different stability conditions at the ARM site and Southern Plains wind farm.

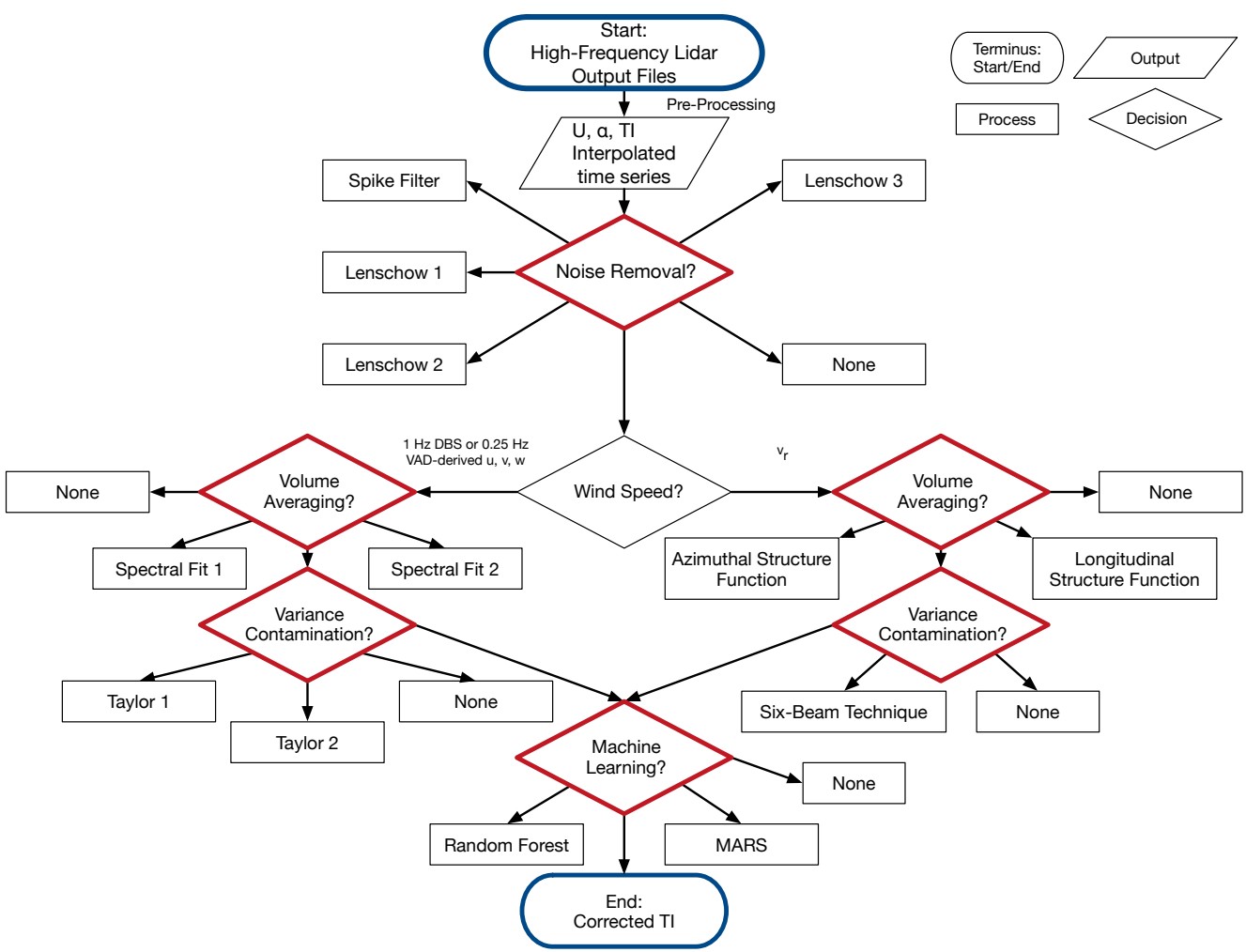

**Figure 1.** Flowchart depicting different methods for correcting TI with L-TERRA. Starting and ending points are indicated by blue-outlined ovals and modules are indicated by red-outlined diamonds.

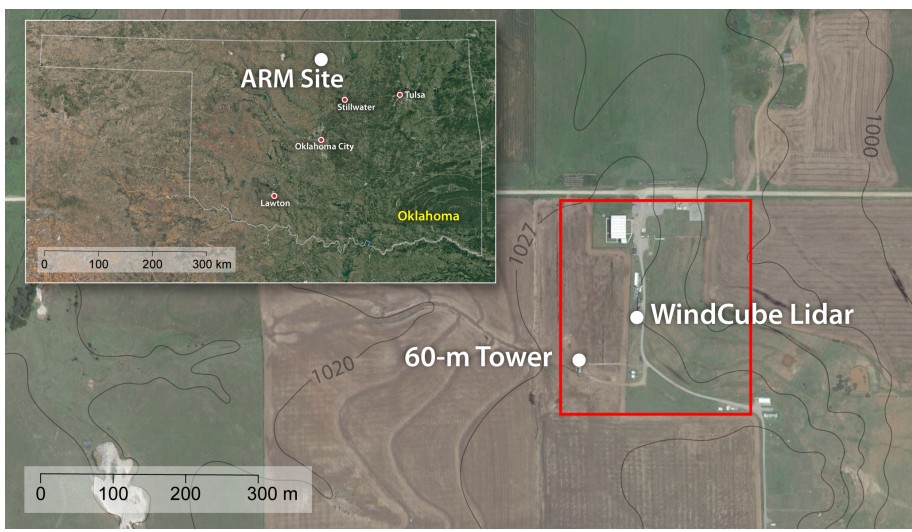

**Figure 2.** Inset: Google Earth image of the state of Oklahoma. Location of Southern Great Plains ARM site is denoted by white marker. Larger figure: Google Earth image of the central facility of the Southern Great Plains ARM site (outlined in red box) with overlaid elevation contours in feet. Elevation map is from the United States Geological Survey and uses contour intervals of approximately 10 feet (3.05 m). Locations of WC lidar and 60m tower are indicated by white markers.

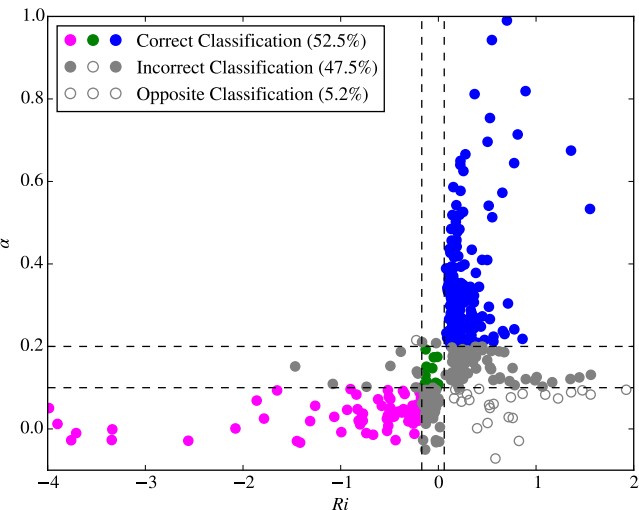

**Figure 3.** Richardson number from tower data at the ARM site vs. shear exponent calculated from WC data. Dashed lines denote stability thresholds as defined in the text. Magenta, green, and blue circles correspond to times when the classification based on both $Ri$ and $\alpha$ was unstable, neutral, or stable, respectively, and gray circles correspond to times when the classification was different. Open gray circles denote times when the classification was stable based on $Ri$ and unstable based on $\alpha$, or vice versa. Percentages of the total combined $Ri$-$\alpha$ dataset corresponding to each case are shown in parentheses in figure legend.

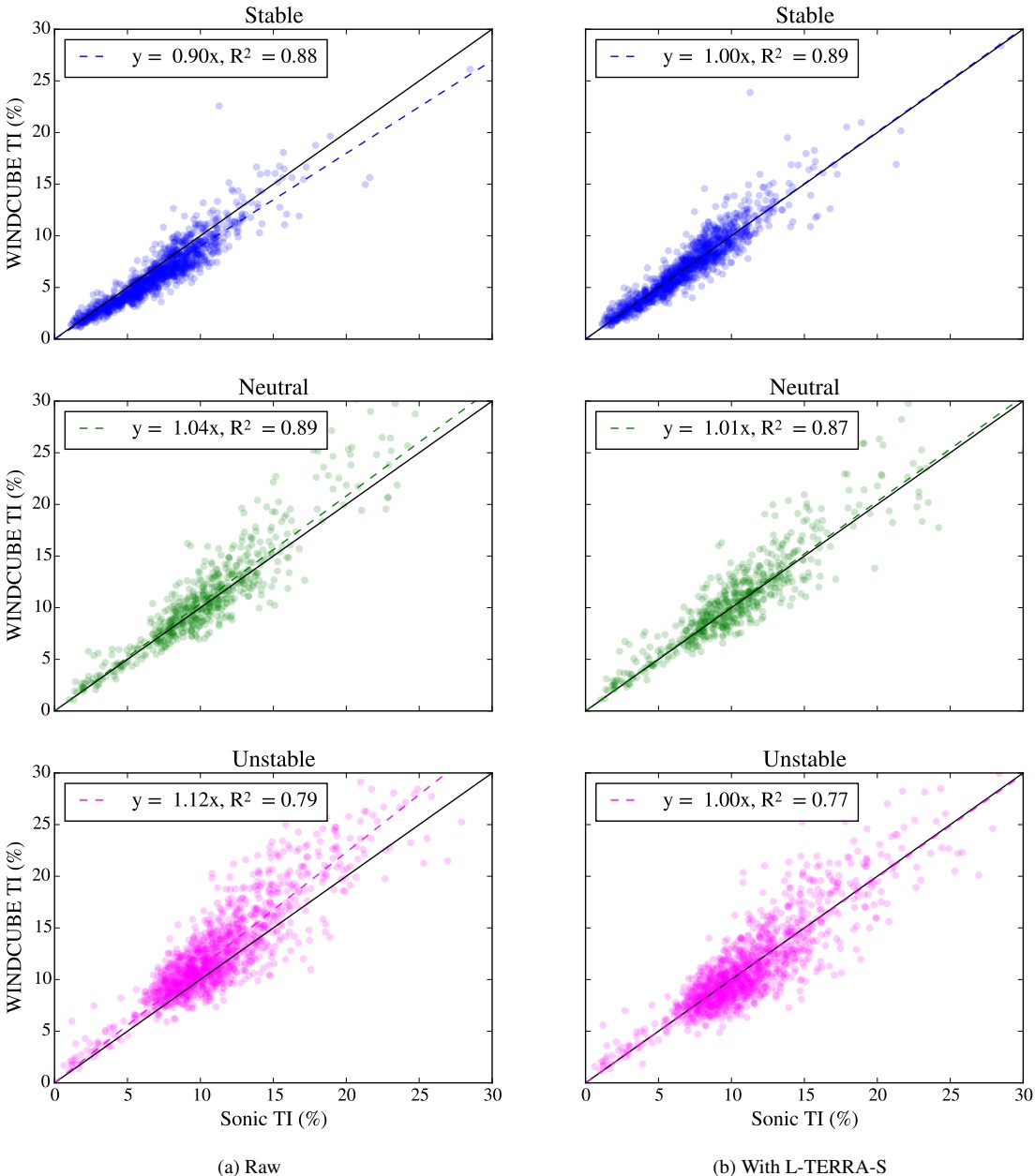

(a) Raw

(b) With L-TERRA-S

**Figure 4.** Scatter plots of met tower vs. WC TI for data from 60m measurement height at the ARM site a) before and b) after L-TERRA-S has been applied. One-to-one line and regression lines are shown for reference and regression line statistics are shown in figure legends.

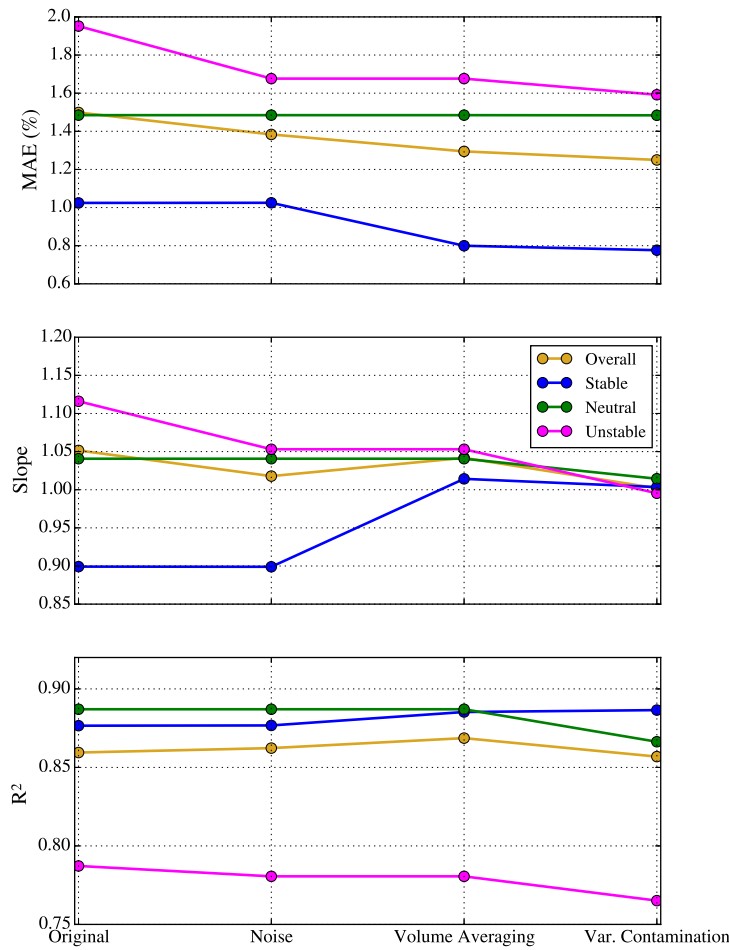

**Figure 5.** Progression of MAE (top), regression line slope (middle), and $R^2$ value of regression line (bottom) for WC vs. sonic TI at the ARM site after application of different modules in L-TERRA-S.

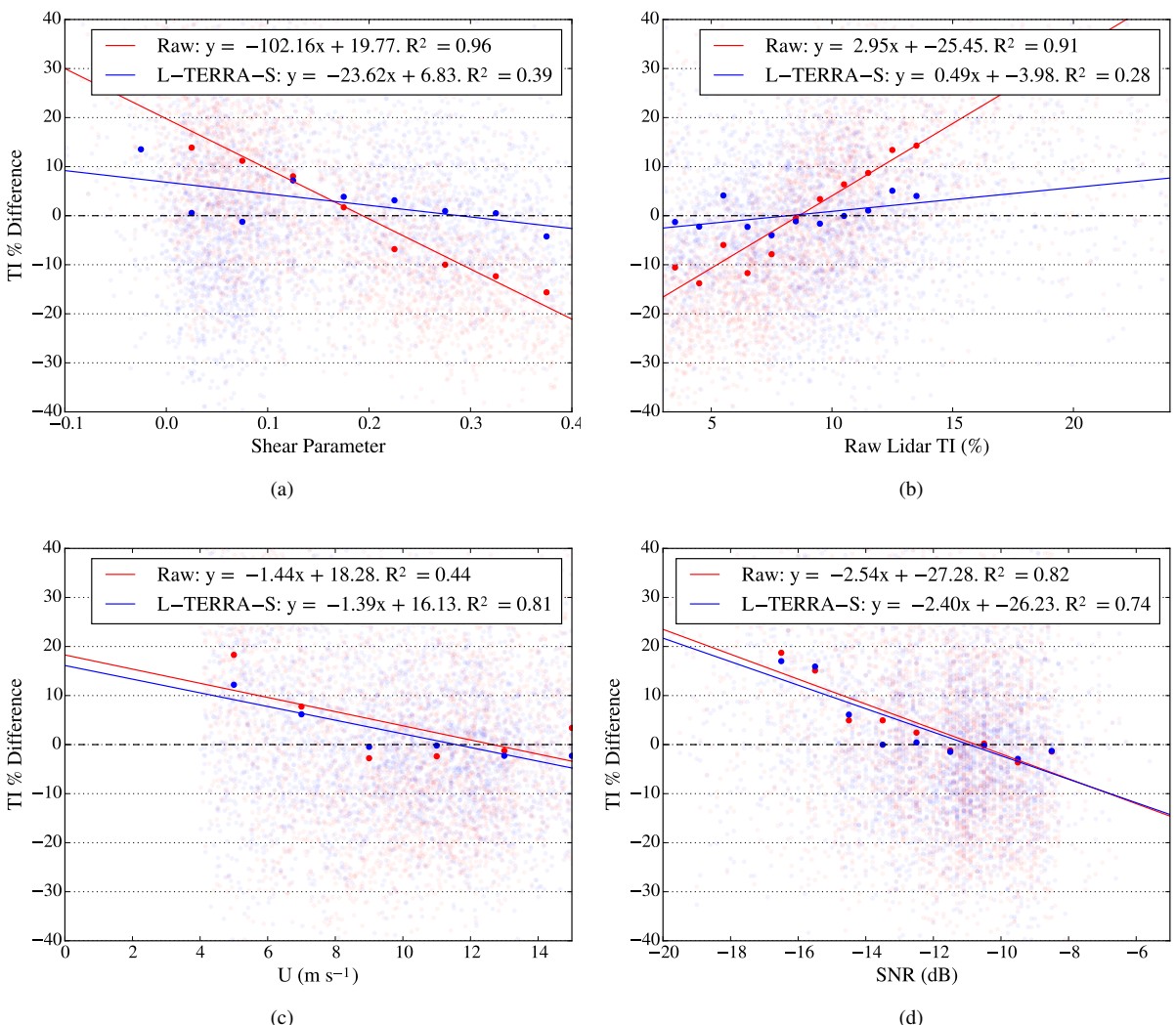

**Figure 6.** Percent difference between WC and sonic TI for the ARM site as a function of a) shear parameter b) raw WC TI c) mean wind speed and d) SNR. Differences are shown both before (red circles) and after (blue circles) L-TERRA-S has been applied. Solid circles correspond to averages of binned data and solid lines correspond to regression line fits to bin-means, following the procedure in Annex L of IEC 61400-12-1, Draft Edition (International Electrotechnical Commission, 2013).

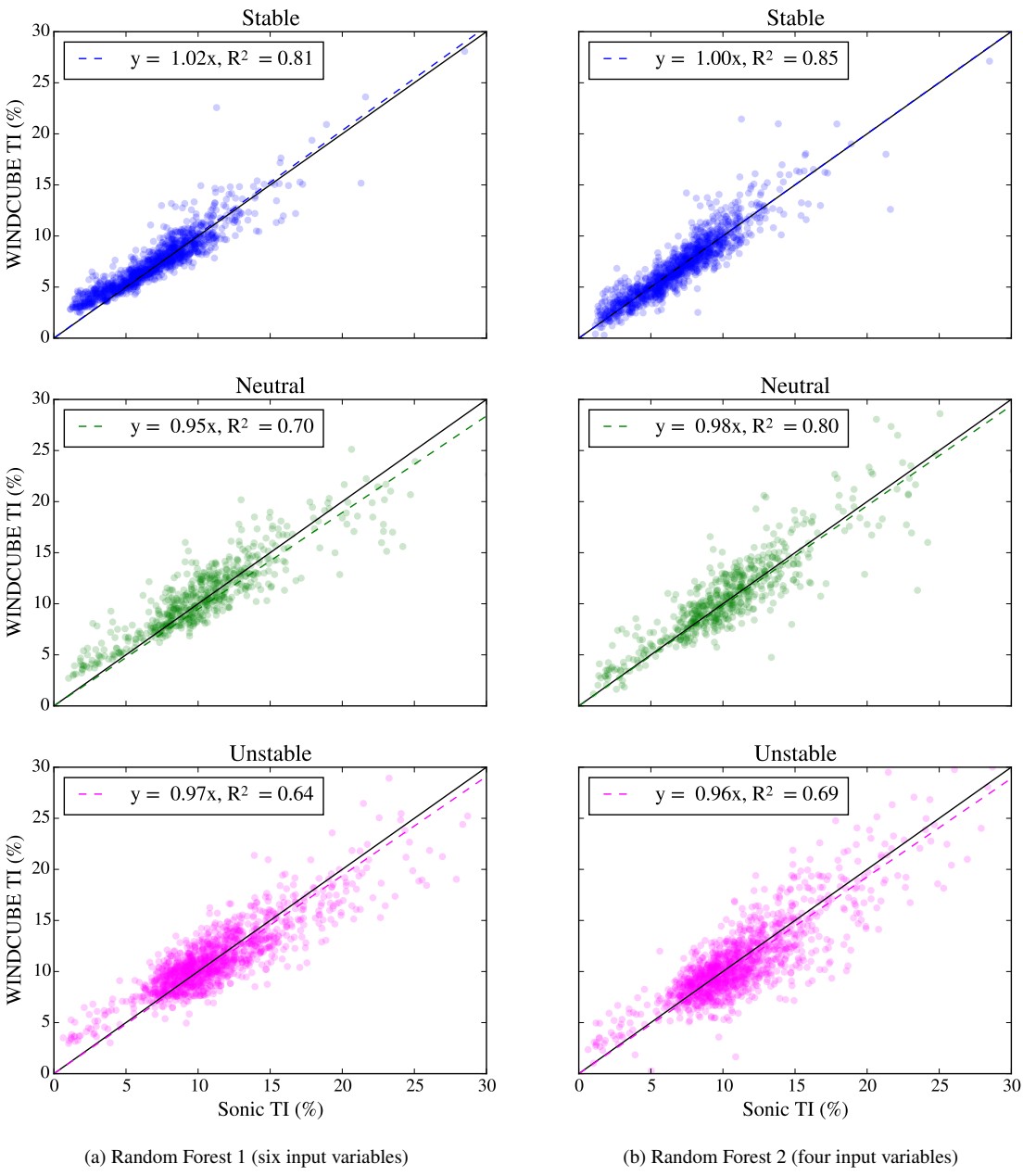

(a) Random Forest 1 (six input variables)     (b) Random Forest 2 (four input variables)

**Figure 7.** Scatter plots of met tower vs. WC TI for data from 60m measurement height at the ARM site a) after application of the first random forest described in the text and b) after application of the second random forest. One-to-one line and regression lines are shown for reference.