# Peer review of "An Error Reduction Algorithm to Improve Lidar Turbulence Estimates for Wind Energy"

_Wind Energy Science, 2016_

## Short Comment (SC1) · 11 Jul 2016

Dear authors,

I got the opportunity to read a small piece of your work by accident and I found something that is not correct in your statements. In Section 2.3.1 (when discussing how to correct lidar turbulence) you state: "Values for these parameters can be estimated by using high-frequency sonic anemometer data, but cannot be obtained by the lidar itself".

If you read the report by Sathe et al. (2015a) you realize that you can predict the radial velocity spectra of a lidar for a given set of turbulence parameters (like the Mann model ones). This means that you can do the inverse operation, i.e. you can predict the turbulence parameters for a given lidar-based radial velocity spectra. And if you are

able to predict these parameters, you can also predict the "ideal" sonic spectra and so estimate the ratio between the sonic and lidar-based variances.

Another point is that it is misleading to talk throughout the manuscript about "correcting lidar turbulence". This sounds like the lidar turbulence was wrong. It is not wrong, it is simply not the same turbulence you expect measuring with a sonic. Perhaps it should always be called lidar-turbulence.

Regards!

––––––––––––––––––––––––––––––

---

## Author Comment (AC1) · 15 Jul 2016

Dear Alfredo,

Thank you for your comments on the manuscript. Regarding your statements on Section 2.3.1, I agree that you can also retrieve parameters for the Mann model from lidar measurements, although there is much more uncertainty in these retrieved parameters. As discussed in Sathe et al. (2015a), the Mann model is still only an approximation of the turbulence field and is only valid under neutral conditions and in flat terrain. Thus, there are still drawbacks to using the Mann model to improve lidar turbulence estimates, and the goal of this work is to improve lidar turbulence estimates under all stability conditions and terrain types. This section has been revised to clarify the limited use cases for the Mann model.

[Figure]

Your point on the phrase "correcting lidar turbulence" is well-taken. It is true that what the lidar is measuring is not incorrect, it is just not the same value of turbulence that would be measured with a sonic anemometer. In the first paragraph of Section 2, it is now stated that "correcting" lidar turbulence refers to techniques that are used to bring lidar turbulence estimates closer to estimates from a cup or sonic anemometer.

Sincerely,

Jennifer Newman

---

## Short Comment (SC2) · 19 Jul 2016

Dear Jennifer,

I think there is still an issue. The idea of my comment was to point out that you can theoretically predict how the radial velocity spectra of the lidar is by knowing 1) the lidar probe volume 2) the velocity tensor and 3) the scanning configuration. This means that this is in principle the way to estimate the difference between variances measured by a sonic and a lidar if they are computed from "single" radial velocity measurements. The issue is that the 2) is not that trivial and the Mann model is just an approximation to the problem (you can always try another model for the velocity tensor). But your discussion should not be centered on the drawbacks of the Mann model but perhaps on the difficulties of the 2). Hope you understand this point.

[Figure]

FYI: the Mann model has been used to predict the spectra on different atmospheric conditions, heights, wind speeds (see Peña et al. 2010) and for different terrain types (see the documentation of WAsP Engineering and the work by Chougule et al., 2014).

And related to the "correcting lidar turbulence": you can clarify that in a sentence but in my opinion I think the term is just incorrect. The reader will still think that it is wrong. Why not simply state always something like "lidar-based turbulence" or something similar?

Regards, Alfredo

Peña et al. (2010) On the length-scale of the wind profile. QJRMS: 136:2119-2131 Chougule et al. (2014) Spectral tensor parameters for wind turbine load modeling from forested and agricultural landscapes. Wind Energy: 18:469-481

---

## Referee Comment (RC1) · Anonymous Referee #1 · 22 Jul 2016

The manuscript states that it presents a new method of estimating turbulence for lidar and as such represents a new contribution. Overall, it is difficult to determine how important this technique is because as far as it has been possible to determine the corrected turbulence is not evaluated/presented except after being filtered through power prediction. These improvements do seem to be very small – can you comment on whether this is significant? And if so which part of the model streams are necessary?

Much of the introductory material is qualitative and could be much improved and likely made briefer by using quantitative metrics. So for example from the abstract it is not possible to evaluate how good the new method is or how much effort is needed to implement it.

This conversational style continues through the Introduction which could be much reduced or improved. For example, 'turbulence' is not defined – so when comparing turbulence from cup anemometers or sonic anemometers versus lidar what is really being compared?

In the Background section there needs to be some reorganization. If there is going to be a general introduction to lidar technology this is where different lidar systems should be compared – with a focus on how 'turbulence' is derived by different systems.

The section on errors in lidar data is probably too broad and needs to be quantitative. Which of these errors will dominate wind speed and turbulence estimation? If not maybe it could simply be a table with references. The WC scanning circle (p4, l5-10) is surely dependent on the instrument and not a general quantity.

Similarly, the section on correcting lidar turbulence gives a qualitative overview. If this cannot be quantitative, which would be best approach, then a table giving the necessary inputs, output and advantages and disadvantages would be helpful.

In section 7 it is not clear why the step with the power law is needed – this surely introduces much larger errors than can be corrected for later?

Is it correct then that you are defining turbulence as the standard deviation of u/u for lidar, cups and sonics? (p7, l26). If so probably best to state this upfront. You say the process is similar for all three but it can't be – no coordinate rotation for cups?

It is my understanding of what you have written that you apply corrections for 1) instrument noise in the form of a spike filter 2) volume averaging and 3) variance contamination and then use machine learning to train (TI error?) on predictor variables including shear parameter, mean wind speed etc.

The datasets are∼6 months from ARM site cf 60 m tower data, 2 months from BAO vs 300 m tower and 7 months from a wind farm.

You had sonic data and used the shear parameter to classify stability, why ? (Table 2)?

[Figure]

Table 3/4 is unclear what it is showing and what has been done? What is the MARS model ? is this different from Terra?

Please work on these Table and Figure captions. It is really difficult to understand what they pertain to. Figure 1: Not sure why this is here or why it is needed. Figure 3/4. Quality needs to be improved. Figure 5. Not sure why these histograms are needed. Which instruments are these from? Figure 6. Not sure why these graphs are needed. What is the corrected turbulence here? Figure 7. Not really sure why the regression lines are needed. Are these the TI corrected data?

I was surprised when I got to the end of the Tables and Figures that no results are presented for the turbulence-estimates given that is the main topic of the paper?

Section 5 is descriptive of the general behavior of turbulence at the sites – is it needed? There did not seem to be anything here that was unexpected so it was not clear why it is present. When I look at Figure 7 I think you are showing that there is better agreement between the sonic TI and the (uncorrected?) lidar TI when the wind shear is lower?

The results section is very unclear. A better approach would be to indicate how the results were obtained rather than stating 'optimal model combinations are shown in Table 3' – this table refers to MAE in kW before and after (training with Terra?) and indicates a reduction in MAE – so not turbulence? Where are the turbulence results? It makes me wonder if I am missing the point of the paper. How important is this? If the overall power is 2MW – then is a reduction in MAE from 2.16 to 1.77 kW important? Actually now I read this I realise I don't follow this at all. Is this like a reduction in error from 0.14% to 0.11% - is it worth this effort? Where does the error in reduction come from? Is this on the average power over one year or ? Unfortunately I was unable to determine the results that lead to the statement 'L-TERRA improves TI estimates' – how would I see that?

Overall the paper had an interesting premise but it needs a major overall to clarify and show the results pertaining to turbulence and to remove unnecessary material. The

error reduction approach has to be systematically compared to other methods and to show the effort level required to obtain it to be of utility. As it stands it would not be possible for anyone to reproduce this work.
* * *

---

## Referee Comment (RC2) · Anonymous Referee #2 · 29 Jul 2016

General comments:

The paper present the L-TERRA algorithm developed for correcting lidar measured turbulence intensity. Correcting here implies retrieving the same TI as that measured by a sonic anemometer. The algorithm contains two main parts:

1. Correction based on physical considerations: correction for the volume averaging effect and correction for the beam cross contamination effect. Those effects are corrected with methods previously developed in other pieces of work.

2. Correction of the reaming error after correction with the physical considerations, using machine learning models. This second part is to my understanding the main novelty of the piece of work presented in the paper.

The first part of the paper is rather well organized and clear:

Section 2 provides a complete review of the work achieved on the topic of turbulence lidar measurement and suggested solutions for correction. Section 3 gives a high level overview of the different modules of the L-TERRA algorithm, although some part should include some more information. Section 4 gives an overview of the three datasets forming the basis of the analysis presented in the paper. Section 5 demonstrate the relevance of accurate TI measurement for wind turbine power performance estimation; a fortiori AEP estimate and resource assessment. An interesting point highlighted in this section is that most of the significant errors in power estimation occur for small errors in TI. The second part, presenting the actual work and results, would gain very much by following a more systematic approach.

The following questions should be answered in the paper:

1. How much of the correction is done by the physical modules and how much by the machine learning module?

2. What is the remaining error, after correction by the physical modules, due to? Is there any systematic pattern in the remaining error? If not (in extreme case, if the remaining error looks random) can a machine learning approach really make an improvement?

3. To which extent the machine learning module can improve the correction from the physical correction modules? In other words, can the improvement compared to previous work cited in section 3 be quantified and clearly demonstrated?

4. In the introduction, the L-TERRA algorithm is presented as a method easier to apply than other methods for lidar turbulence measurement correction proposed so far (p2, l25). However, in the end, the method does not look really much easier (and maybe even more complex) to implement since it re-uses or adapts the methods qualified as complex to the WC, combine them and add the machine learning module. The

simplicity of the approach needs to be further demonstrated.

Furthermore, a more straight forward evaluation of the results, like a direct comparison of the lidar measured turbulence intensity to the sonic measurements before and after correction (as done in figure 11) would lead to clearer conclusions. The conversion of the TI/TI error into power/power error is very interesting to demonstrate the impact of the TI measurement error on power prediction (as done in section 5), but it tends to confuse the analysis objectives when it is used in the results (like it is now done in section 6.2.).

Detailed comments:

(p: page; l: line)

P2; l.27-32: this paragraph is misleading and/or misplaced. It gives the conclusions of the analysis. I suggest removing it.

P7, l5: is it 4s second scan or 5 sec scan?

P8, l7: "Two methods were evaluated. . ." The flow chart in fig. 2 shows 4 methods to take care of the effect of volume averaging (2 for each type of wind speed). Does that mean only 2 of them have been tested? Which ones: "spectral filtering 1" and "spectral filtering 2"?

P8, l11-12: could you please provide explanations on how you have applied the method from Krishnamurthy to DBS scans?

P8, l17-19: could you please provide explanations on how you have applied those techniques to a 5 beam configuration?

P8, l24: "there is still some error . . ." Why? What is the remaining error due to? Does it mean the physical correction models applied previously are not good enough? The assumptions are not verified?

P9, l1: for the paper to be comprehensive by itself, it would be good to have a short

description of each of the 3 machine learning methods. P9, l21-22:

1. Some of these results are rather surprising to me: internal temperature and pitch. Could you please comment on those and provide some information regarding the range of each variable and its correlation with the TI?

2. TI and sigma_w were not correlated?

3. Have you performed the same sensitivity analysis for the other flat site to see if you get the same final predictor variables?

P10, section 4 1. In the introduction you mention 2 flat sites and one semi-complex. Which one of the three sites described here is the semi-complex one? My guess is BAO, but needs to be clearly sated as it can influence the results of the lidar measurements. 2. In the semi complex site, were the lidar measurements corrected for the effect of complex terrain (e.g. FCR) or the same wind speed reconstruction algorithm was used at all sites?

P12, section 5 The main reason for the poor results at BAO is probably the effect of the terrain on the flow. In complex terrain, the assumption of horizontal homogeneity is not verified, then the reconstructed mean wind speed includes some error and therefore does not correlate with the sonic measurements as well as lidar measurements in flat terrain. If the mean wind speed comparison to the sonic is poor, the TI comparison is also expected to be poor.

P13, l19-20: Does this mean that the correction algorithm for the radial wind speed ( right part of the flow chart In figure 2) has not been tested yet (or at least the results are not included in this paper)? This sounds in contradiction with p13, l22-23 (and p6, l29-30) stating that "all possible combinations . . . were evaluated". Could you please clarify which of the process presented in flow chart in fig 2 have actually been used for the results presented and discussed in this paper?

P13, l 30-31: Only the optimal combinations for each site are presented in table 3. 1.

How much difference was there between the different combinations? Was it significant? 2. Is one of the combinations more robust than the others? i.e. could you , based on this analysis, recommend one combination or is the idea to always try all of them and pick the smallest error? (This is maybe to be included in the discussion in section 6).

p.14, l13-28: 1. From this analysis, it sounds like the cross contamination effect is the most important. How much the correction for this effect change the lidar does measured TI? And how much does the machine learning correction part change the lidar measured TI?

2. Figure 11 shows that the slopes in the linear regression are usually getting closer to 1 after application of the L-TERRA correction, which mean the mean error is reduced. But the scatter is increased ($R^2$ is lower), so the improvement is actually mitigated. Moreover, this shows that the method does not necessarily provide better estimate of every 10 minute value of TI measure by the lidar, whereas it was demonstrated in section 5 that is what was needed.

P15, section 6.2: only MARS and RF are discussed. Support Vector Regression was not tested?

---

## Author Response (AR1)

Response to Reviewer 1

Dear Reviewer,

Thank you for your thorough review of the manuscript. In response to your comments, we have substantially revised the manuscript to make it more clear and focused. The majority of the changes have been made to the results section, where we now focus on showing improvements in lidar TI, rather than improvements in power prediction. We have also reduced the number of figures and aimed to make figure and table captions more descriptive. Another major change is the removal of the BAO dataset from the manuscript, as we did not feel confident in the quality of the lidar data from the site. A point-by-point response to your comments is given below.
* * *
*The manuscript states that it presents a new method of estimating turbulence for lidar and as such represents a new contribution. Overall, it is difficult to determine how important this technique is because as far as it has been possible to determine the corrected turbulence is not evaluated/presented except after being filtered through power prediction. These improvements do seem to be very small – can you comment on whether this is significant? And if so which part of the model streams are necessary?*

**Response:** Results from the TI correction are now shown in Section 5 of the revised manuscript. Several metrics are given to indicate improvement in the lidar TI, including MAE, regression line slope, and changes in TI error sensitivity to different external conditions. It was determined that different parts of the model stream are needed depending on the stability conditions, which is discussed on p. 11, Lines 16-21.

*Much of the introductory material is qualitative and could be much improved and likely made briefer by using quantitative metrics. So for example from the abstract it is not possible to evaluate how good the new method is or how much effort is needed to implement it.*

**Response:** We have now included a quantitative indication of the improvement of lidar TI with L-TERRA in the abstract and have also given this information in Tables 3 and 4 in the text.

*This conversational style continues through the Introduction which could be much reduced or improved. For example, 'turbulence' is not defined – so when comparing turbulence from cup anemometers or sonic anemometers versus lidar what is really being compared?*

**Response:** Turbulence intensity has now been defined in the introduction section. We also acknowledge the difference in how TI is calculated for cup vs. sonic anemometers in Section 3.1 (p. 8, Lines 7-11).

*In the Background section there needs to be some reorganization. If there is going to be a general introduction to lidar technology this is where different lidar systems should be compared – with a focus on how 'turbulence' is derived by different systems.*

**Response:** The background section has been revised. Information on different lidar systems and methods for deriving turbulence is now included in Section 2.1 – Lidar Technology.

*The section on errors in lidar data is probably too broad and needs to be quantitative. Which of these errors will dominate wind speed and turbulence estimation? If not maybe it could simply be a table with references. The WC scanning circle (p4, l5-10) is surely dependent on the instrument and not a general quantity.*

**Response:** As our goal was to inform the reader qualitatively about the different sources of error, we have decided to keep this section intact. We agree that it's important to determine which sources of error will dominate wind speed and turbulence estimation, but this is a complex question, as it depends on atmospheric conditions, site conditions, and the lidar technology and scanning strategy used.

*Similarly, the section on correcting lidar turbulence gives a qualitative overview. If this cannot be quantitative, which would be best approach, then a table giving the necessary inputs, output and advantages and disadvantages would be helpful.*

**Response:** We have decided to keep this information in the text, as the manuscript already contains five tables and much of the information on how the techniques are implemented cannot be fit into a table. We appreciate the suggestion.

*In section 7 it is not clear why the step with the power law is needed – this surely introduces much larger errors than can be corrected for later?*

**Response:** The power law is used to determine the shear parameter α, which is then used as a proxy for atmospheric stability and as an input variable for the machine learning models discussed in Section 5.2. Although there are certainly limitations to using the power law, our goal is merely to relate what the lidar is measuring to the errors in the TI.

*Is it correct then that you are defining turbulence as the standard deviation of u/u for lidar, cups and sonics? (p7, l26). If so probably best to state this upfront. You say the process is similar for all three but it can't be – no coordinate rotation for cups?*

**Response:** It has now been stated in Section 3.1 (p. 8, Lines 7-11) that the way that TI is calculated for cup anemometers is different than how it is calculated for sonic and lidar measurements.

*It is my understanding of what you have written that you apply corrections for 1) instrument noise in the form of a spike filter 2) volume averaging and 3) variance contamination and then use machine learning to train (TI error?) on predictor variables including shear parameter, mean wind speed etc.*

**Response:** This is correct and is explained in Figure 2 of the manuscript.

*The datasets are   6 months from ARM site cf 60 m tower data, 2 months from BAO vs 300 m tower and 7 months from a wind farm.*

**Response:** The BAO dataset has been removed from the manuscript, so we now use approximately 6 months of data from both the ARM site and the wind farm.

*You had sonic data and used the shear parameter to classify stability, why? (Table 2)?*

**Response:** We agree that using sonic anemometer data to calculate the Monin-Obukhov length or Richardson number would be a more ideal way to classify stability. However, we wanted to use information from a stand-alone lidar to classify stability, which is why we decided to use the shear parameter as a proxy for stability (p. 10, Lines 23-25).

*Table 3/4 is unclear what it is showing and what has been done? What is the MARS model? is this different from Terra?*

**Response:** The information in Table 3 has now been split into three tables in the revised manuscript: Table 5 contains optimal model combinations for L-TERRA, and Tables 3 and 4 contain TI error metrics for the ARM site and wind farm, respectively. In addition, the physics-based corrections and machine-learning corrections are now discussed in two separate sections (Sections 5.1 and 5.2).

*Please work on these Table and Figure captions. It is really difficult to understand what they pertain to.*

**Response:** The table and figure captions have been revised.

*Figure 1: Not sure why this is here or why it is needed.*

**Response:**  Figure 1b (power sensitivity to TI) has now been removed from the manuscript. Figure 1a (TI-dependent power curves) has been retained to indicate the importance of measuring TI to determine turbine power production.

*Figure 3/4. Quality needs to be improved.*

**Response:** The quality of Figure 3 has been improved. Figure 4 has been removed from the manuscript, as the BAO data are now longer discussed.

*Figure 5. Not sure why these histograms are needed. Which instruments are these from?*

**Response:** This figure has been removed from the manuscript.

*Figure 6. Not sure why these graphs are needed. What is the corrected turbulence here?*

**Response:** This figure has been removed from the manuscript.

*Figure 7. Not really sure why the regression lines are needed. Are these the TI corrected data?*

**Response:** This figure has been removed from the manuscript. The raw and corrected TI for the ARM site are now shown in Figure 4. To make the figures more clear, the number of stability classes has been reduced to three, and different plots are used to show TI data for different stability classes.

*I was surprised when I got to the end of the Tables and Figures that no results are presented for the turbulence-estimates given that is the main topic of the paper?*

**Response:** We agree that results for the turbulence estimates should be shown in the paper. We now focus solely on improvement of the lidar TI and plan to discuss applications of the TI correction (e.g., on power prediction) in future work.

*Section 5 is descriptive of the general behavior of turbulence at the sites – is it needed?\ There did not seem to be anything here that was unexpected so it was not clear why it is present. When I look at Figure 7 I think you are showing that there is better agreement between the sonic TI and the (uncorrected?) lidar TI when the wind shear is lower?*

**Response:** Section 5 has now been removed from the manuscript.

*The results section is very unclear. A better approach would be to indicate how the results were obtained rather than stating 'optimal model combinations are shown in Table 3' – this table refers to MAE in kW before and after (training with Terra?) and indicates a reduction in MAE – so not turbulence? Where are the turbulence results? It makes me wonder if I am missing the point of the paper. How important is this? If the overall power is 2MW – then is a reduction in MAE from 2.16 to 1.77 kW important? Actually now I read this I realise I don't follow this at all. Is this like a reduction in error from 0.14% to 0.11% - is it worth this effort? Where does the error in reduction come from? Is this on the average power over one year or ? Unfortunately I was unable to determine the results that lead to the statement 'L-TERRA improves TI estimates' – how would I see that?*

**Response:** We now focus only on the results of the turbulence correction and do not translate the improvements in TI to improvements in power prediction. We agree that this translation to power prediction made the results section unclear. In addition, we now show changes in TI error sensitivity that result from application of L-TERRA (Figure 5), as the reduction of TI error sensitivity to external parameters is another way to demonstrate the benefits of L-TERRA.

*Overall the paper had an interesting premise but it needs a major overall to clarify and show the results pertaining to turbulence and to remove unnecessary material. The error reduction approach has to be systematically compared to other methods and to show the effort level required to obtain it to be of utility. As it stands it would not be possible for anyone to reproduce this work.*

**Response:** We have significantly revised the manuscript to clarify the techniques used and focus solely on the correction of lidar turbulence. It would be difficult to compare the results of L-TERRA quantitatively, as most of the current turbulence error reduction techniques discussed in the manuscript use scanning lidars or data that is not readily available from vertically profiling lidars. However, we greatly appreciate your suggestions to make the manuscript more clear and concise.

Response to Reviewer 2

Dear Reviewer,

Thank you for your thorough and extremely helpful review of the manuscript. In response to your comments, we have substantially revised the manuscript and focused on improving the lidar TI, rather than reducing errors in power prediction. A more complete description of the physics-based corrections and machine learning techniques is now included, as well as a thorough and clear discussion of the results from both the physics-based and machine learning corrections in L-TERRA.  A point-by-point response to your comments is given below.
* * *
*The following questions should be answered in the paper:*

*1. How much of the correction is done by the physical modules and how much by the machine learning module?*

**Response:**  The application of the physics-based corrections and the machine learning techniques are now separated into two different sections (Sections 5.1 and 5.2), so the effects of each module on the resulting TI estimates can now be seen clearly.

*2. What is the remaining error, after correction by the physical modules, due to? Is there any systematic pattern in the remaining error? If not (in extreme case, if the remaining error looks random) can a machine learning approach really make an improvement?*

**Response:** A sensitivity analysis was conducted in Section 5.2, and the use of the sensitivity analysis results to determine patterns in the remaining error is discussed on p. 12, Lines 24-30, and p. 13, Lines 32-25–p. 14, Lines 1-4.

*3. To which extent the machine learning module can improve the correction from the physical correction modules? In other words, can the improvement compared to previous work cited in section 3 be quantified and clearly demonstrated?*

**Response:** The physical corrections and machine learning methods are now discussed separately in the revised manuscript.  It was determined that training a random forest or MARS model on one dataset and testing on a different dataset led to an increase in MAE and decrease in $R^2$ values. While training and testing on the same site did decrease MAE values, it still increased $R^2$ values.

It would indeed be useful to compare our approach to the results from previous methods, but it would be difficult to make this comparison in reality.  Most of the

previous methods discussed require information that is not readily available from a vertically profiling lidar.

*4. In the introduction, the L-TERRA algorithm is presented as a method easier to apply than other methods for lidar turbulence measurement correction proposed so far (p2, 25). However, in the end, the method does not look really much easier (and maybe even more complex) to implement since it re-uses or adapts the methods qualified as complex to the WC, combine them and add the machine learning module. The simplicity of the approach needs to be further demonstrated.*

**Response:** Section 3.6 (pp. 9-10) has been added to compare the evaluation of L-TERRA to other methods described in the text.

*Furthermore, a more straight forward evaluation of the results, like a direct comparison of the lidar measured turbulence intensity to the sonic measurements before and after correction (as done in figure 11) would lead to clearer conclusions. The conversion of the TI/TI error into power/power error is very interesting to demonstrate the impact of the TI measurement error on power prediction (as done in section 5), but it tends to confuse the analysis objectives when it is used in the results (like it is now done in section 6.2.).*

**Response:** In the revised manuscript, direct comparisons of the TI have been made both before and after L-TERRA has been applied (Section 5). To make the paper more clear, we have decided to reserve the implications of L-TERRA on power prediction for future work.

*Detailed comments:*
*(p: page; l: line)*

*P2; l.27-32: this paragraph is misleading and/or misplaced. It gives the conclusions of the analysis. I suggest removing it.*

**Response:** This paragraph has been removed.

*P7, l5: is it 4s second scan or 5 sec scan?*

**Response:** The actual accumulation time of the WINDCUBE lidar used in this work was just under 1 second, so a full scan actually took closer to 4 seconds. This has been clarified in the revised manuscript.

*P8, l7: "Two methods were evaluated. . ." The flow chart in fig. 2 shows 4 methods to take care of the effect of volume averaging (2 for each type of wind speed). Does that mean only 2 of them have been tested? Which ones: "spectral filtering 1" and "spectral filtering 2"?*

**Response:** It has now been clarified that only the model path that incorporates the *u, v*, and *w* components was tested, as not all vertically profiling lidars include the radial velocity measurements in their output files (p. 7, Lines 7-10). The difference between Spectral Fit 1 and Spectral Fit 2 has also now been clarified in the manuscript (p. 9, Lines 1-4).

*P8, l11-12: could you please provide explanations on how you have applied the method from Krishnamurthy to DBS scans?*

**Response:** A description of applying the structure function technique to DBS data is now described in Section 3.3 (p. 8, Lines 22-26).

*P8, l17-19: could you please provide explanations on how you have applied those techniques to a 5 beam configuration?*

**Response:** An explanation for applying the six-beam technique to a five-beam configuration has been added to Section 3.4 (p. 9, Lines 8-12).

*P8, l24: "there is still some error . . ." Why? What is the remaining error due to? Does it mean the physical correction models applied previously are not good enough? The assumptions are not verified?*

**Response:** This section has been revised so that it now simply describes the content of the machine learning module in L-TERRA (Section 3.5).

*P9, l1: for the paper to be comprehensive by itself, it would be good to have a short description of each of the 3 machine learning methods.*

**Response:** Random forests and the MARS model are now described in more detail in the revised manuscript (p. 9, Lines 15-22). References to support vector regression were removed in the revised manuscript, as this model performed poorly on the test datasets in comparison to MARS and random forests.

*P9, l21-22:*
*1. Some of these results are rather surprising to me: internal temperature and pitch. Could you please comment on those and provide some information regarding the range of each variable and its correlation with the TI?*

**Response:** We agree that the high sensitivity of TI error to internal temperature and pitch was a bit surprising. After applying the new version of L-TERRA in the revised version of the paper, this high sensitivity to internal temperature and pitch was no longer evident.

*2. TI and sigma_w were not correlated?*

**Response:** TI and sigma_w were correlated, though not extremely strongly (correlation coefficient = 0.363). We used a correlation coefficient of 0.5 to discriminate between weakly and strongly correlated variables, so both TI and sigma_w were retained in the list of predictors.

*3. Have you performed the same sensitivity analysis for the other flat site to see if you get the same final predictor variables?*

**Response:** We did perform the same sensitivity analysis for the other flat site and determined that while some variables had similar sensitivities at both sites, others had significantly different sensitivities. This is discussed on p. 13, Lines 6-10 in the revised manuscript.

*P10, section 4*

*1. In the introduction you mention 2 flat sites and one semi-complex.*
*Which one of the three sites described here is the semi-complex one? My guess is BAO, but needs to be clearly sated as it can influence the results of the lidar measurements.*

**Response:** The BAO was indeed the semi-complex site. References to the BAO have been removed from the revised manuscript, as we were not confident in the lidar data quality from the site.

*2. In the semi complex site, were the lidar measurements corrected for the effect of complex terrain (e.g. FCR) or the same wind speed reconstruction algorithm was used at all sites?*

**Response:** The lidar measurements were not corrected for the effect of complex terrain.

*P12, section 5 The main reason for the poor results at BAO is probably the effect of the terrain on the flow. In complex terrain, the assumption of horizontal homogeneity is not verified, then the reconstructed mean wind speed includes some error and therefore does not correlate with the sonic measurements as well as lidar measurements in flat terrain. If the mean wind speed comparison to the sonic is poor, the TI comparison is also expected to be poor.*

**Response:** We agree that the effect of complex terrain on the flow, in addition to the low aerosol count at the BAO, adversely affected the accuracy of the lidar measurements at the site. Thus, we have removed the BAO data from our analysis.

*P13, l19-20: Does this mean that the correction algorithm for the radial wind speed (right part of the flow chart In figure 2) has not been tested yet (or at least the results are not included in this paper)? This sounds in contradiction with p13, l22-23 (and p6, l29-30) stating that "all possible combinations . . . were evaluated". Could you please clarify which of the process presented in flow chart in fig 2 have actually been used for the results presented and discussed in this paper?*

**Response:** The correction algorithm for radial wind speed has not yet been tested extensively and was not included in this work. This is now stated on p. 7, Lines 7-10 of the revised manuscript.

*P13, l 30-31: Only the optimal combinations for each site are presented in table 3.*

*1. How much difference was there between the different combinations? Was it significant?*

**Response:** Several model combinations produced similar MAE values for each site. This is now stated on p. 11, Lines 9-11.

*2. Is one of the combinations more robust than the others? i.e. could you , based on this analysis, recommend one combination or is the idea to always try all of them and pick the smallest error? (This is maybe to be included in the discussion in section 6).*

**Response:** As minimizing the MAE produced several similar "optimal" model combinations, we suggest that looking at other parameters might be more useful for determining the ideal model combination (p. 11, Lines 11-14). However, for this initial evaluation of L-TERRA, minimizing the MAE is helpful for finding some baseline model combinations for comparison with the raw TI.

*p.14, l13-28:*

*1. From this analysis, it sounds like the cross contamination effect is the most important. How much the correction for this effect change the lidar does measured TI? And how much does the machine learning correction part change the lidar measured TI?*

**Response:** We still believe the cross-contamination effect is important, although we have now phrased the effect in terms of changes in the scales of vertical velocity across the lidar scanning circle (p. 13, Lines 32-35-p. 14, Lines 1-4). For the physics-based corrections, the variance contamination module reduced TI estimates by an average of 0.15% for stable conditions and 0.8% for unstable conditions. As discussed in Section 6.2, machine learning techniques increased MAE and scatter in the TI estimates and likely did not have a large effect on reducing variance contamination. We are currently examining ways to improve the physics-based variance contamination module to further improve TI estimates.

*2. Figure 11 shows that the slopes in the linear regression are usually getting closer to 1 after application of the L-TERRA correction, which mean the mean error is reduced. But the scatter is increased (R^2 is lower), so the improvement is actually mitigated. Moreover, this shows that the method does not necessarily provide better estimate of every 10 minute value of TI measure by the lidar, whereas it was demonstrated in section 5 that is what was needed.*

**Response:** This increase in scatter was mitigated through application of the stability-dependent version of L-TERRA (see Tables 3 and 4 in the revised manuscript). However, we agree that the $R^2$ values still indicate a large amount of scatter in the data and suggest that this scatter occurs because we are not completely capturing all the physics that affect TI error (p. 12, Lines 4-7).

*P15, section 6.2: only MARS and RF are discussed. Support Vector Regression was not tested?*

**Response:** Support vector regression was initially tested but produced poor results in comparison to random forests and the MARS model, so we have not included a discussion of support vector regression in the manuscript. We have also removed references to support vector regression in the section on machine learning (Section 3.5).

**An Error Reduction Algorithm to Improve Lidar Turbulence Estimates for Wind Energy**

Jennifer F. Newman[1] and Andrew Clifton[2]

[1]National Wind Technology Center, National Renewable Energy Laboratory, Golden, CO, 80401, USA
[2]Power Systems Engineering Center, National Renewable Energy Laboratory, Golden, CO, 80401, USA

*Correspondence to:* Jennifer F. Newman (Jennifer.Newman@nrel.gov)

**Abstract.**

Remote sensing devices such as lidars are currently being investigated as alternatives to cup anemometers on meteorological towers. Although lidars can measure mean wind speeds at heights spanning an entire turbine rotor disk and can be easily moved from one location to another, they measure different values of turbulence than an instrument on a tower. Current methods for

5  improving lidar turbulence estimates include the use of analytical turbulence models and expensive scanning lidars. While these methods provide accurate results in a research setting, they cannot be easily applied to smaller,  vertically profiling lidars in locations where high-resolution sonic anemometer data are not available. Thus, there is clearly a need for a turbulence error reduction model that is simpler and more easily applicable to lidars that are used in the wind energy industry.

10  In this work, a new turbulence error reduction algorithm for lidars is described. The algorithm, L-TERRA, can be applied using only data from a stand-alone  vertically profiling lidar and requires minimal training with meteorological tower data. The basis of L-TERRA is a series of physics-based corrections that are applied to the lidar data to mitigate errors from instrument noise, volume averaging, and variance contamination. These corrections are applied in conjunction with a trained machine-learning model to improve turbulence estimates from a vertically profiling WINDCUBE v2

15  lidar. The lessons learned from creating the L-TERRA model for a WINDCUBE v2 lidar can also be applied to other lidar devices.

L-TERRA was tested on data from  two sites in the Southern Plains region of the United States. The physics-based corrections in L-TERRA  reduced the mean absolute error by nearly 20% at both sites and significantly reduced the

20  sensitivity of lidar turbulence errors to atmospheric stability. The accuracy of machine-learning ~~portion of the model was trained on one site and applied to a different site. Errors in turbulence were then related to errors in power through the use of a power prediction model for a simulated 1.5MW turbine. L-TERRA also reduced errors in power significantly at all three sites, although moderate power errors remained for periods when the mean wind speed was close to the rated wind speed of the turbine and periods when variance contamination had a large effect on the lidar turbulence~~ methods in L-TERRA was highly

25  dependent on the input variables and training dataset used, suggesting that machine learning may not be the best technique for

reducing lidar TI error. Future work will include the use of a lidar simulator to better understand how different factors affect lidar turbulence error and to determine how these errors can be reduced using information from a stand-alone lidar.

**1 Introduction**

As turbine hub heights increase and wind energy expands to complex and offshore sites, new measurements of the wind resource are needed to inform decisions about site suitability and turbine selection. Currently, most of these measurements are collected by cup anemometers on meteorological (met) towers. Met towers are fixed in location and typically only collect measurements up to and including the height corresponding to the turbine hub height. However, the measurement of wind speeds across the entire turbine rotor disk is extremely important for power estimation (e.g., Wagner et al., 2009), particularly as modern turbines increase in size. In addition, met towers are expensive to construct and maintain; the estimated cost for installing and maintaining an 80m land-based met tower for a 2-year campaign is €92,000 (≈ 105,000 USD; Boquet et al., 2010). In response to the limitations of met towers for wind energy, remote sensing devices such as lidars (light detection and ranging) have been proposed as potential alternatives to cup anemometers on towers. Lidars are now frequently used in the research community (e.g., Barthelmie et al., 2013; Stawiarski et al., 2013; Fuertes et al., 2014; Sathe et al., 2015b), and acceptance of lidars in the wind energy community is increasing.  The use of remote sensing devices for power performance testing in flat terrain is discussed in Annex L of the most recent draft version of IEC 61400-12-1 (International Electrotechnical Commission, 2013).

While lidars are capable of measuring mean wind speeds at several different measurement heights (e.g., Sjöholm et al., 2008; Peña et al., 2009; Barthelmie et al., 2013; Sathe et al., 2015b), they measure different values of turbulence than a cup or sonic anemometer (e.g., Sathe et al., 2011; Newman et al., 2016b). Turbulence, a measure of small-scale fluctuations in the atmospheric flow, is an extremely important parameter in the wind energy industry. Turbulence measurements are used to classify potential wind farm sites and select suitable turbines (International Electrotechnical Commission, 2005) and can also impact power production. Figure 1 shows the response of a modeled 1.5MW turbine to different 10min mean hub-height wind speeds and levels of hub-height turbulence intensity (TI), defined as the standard deviation of the streamwise wind speed divided by the average wind speed over a 10min period and multiplied by 100%. The power produced by the turbine is profoundly impacted by the level of turbulence, particularly near the rated wind speed of the turbine. Because of the paramount importance of turbulence measurements to the wind energy industry, lidars must be able to accurately measure turbulence to be considered a viable alternative to met towers. The inability of lidars to accurately measure turbulence is currently one of the main barriers to replacing met towers with lidars.

In this work, a new turbulence error reduction model, the Lidar Turbulence Error Reduction Algorithm (L-TERRA), was developed for the WINDCUBE v2 (WC) vertically profiling lidar. The model combines  physics-based corrections, such as a spectral correction, with machine-learning techniques to improve lidar turbulence estimates. While the physics-based corrections can be applied using data from the lidar itself, the machine-learning portion of L-TERRA requires training with a collocated lidar/met

tower dataset. Unlike other methods for improving lidar turbulence estimates, L-TERRA is a simple method that can be easily applied to  vertically profiling lidars. The goal of L-TERRA

5 ~~when the model is trained at one site and applied to data at a different site. Large errors in power prediction still remain for wind speeds near the rated wind speed of the turbine, where the power curve is extremely sensitive to errors in TI , and for low-shear, high-TI conditions, where variance contamination has a strong impact on lidar TI error. Future work will include the use of a lidar simulator to refine the corrections in L-TERRA andexpansion of L-TERRA to different lidar models and configurations~~
[revised manuscript text omitted]

**3.4.1**

~~The previous three modules (instrument noise, volume averaging, and variance contamination) constitute the physics-based corrections of L-TERRA. These modules rely only on data from the lidar itself and use theory rather than mathematical models. While the physies-based corrections do reduce lidar TI error, there is still some error in the lidar TI estimates in comparison to estimates from a cup or sonic anemometer. Thus, machine-learning methods were used in the final step of L-TERRA to bring lidar TI estimates even closer to met tower estimates.~~

**3.5 Machine learning**

 Two machine-learning methods were evaluated as part of L-TERRA: random forests  and multivariate adaptive regression splines (MARS). Random forest models are developed by averaging multiple decision trees that were trained on different subsets of the data. By averaging tens or hundreds of decision trees, the variance of the overall model is reduced significantly (Friedman et al., 2001). Random forests were evaluated because they are relatively easy to understand and have previously been used for wind energy applications (e.g., Clifton et al., 2013; Bulaevskaya et al., 2015).  MARS is essentially a stepwise regression model, where different coefficients and basis functions are used to predict the output depending on each region in the dataset (Friedman, 1991). MARS is wellsuited for the prediction of physical processes .   due to its ability to model non-linearities and interactions among variables.

Potential predictor variables for the machine-learning models were divided into two broad categories: atmospheric state and lidar operating characteristics. Variables that were evaluated as predictor variables in L-TERRA are given in Table 1. Atmospheric state variables included shear parameter, mean wind speed, Doppler spectral broadening, and $u$ and $w$ velocity variances. Lidar operating characteristics included signal-to-noise ratio (SNR) and internal instrument temperature.  In all,  18 predictor variables were considered for the machine learning portion of L-TERRA.

~~Sensitivity of the TI error to the various predictor variables was assessed following the guidelines in Annex L of the new committee draft of IEC 61400-12-1 . Data from the Atmospheric Radiation Measurement (ARM) site, described in Section 4, were used to assess the importance of different predictor variables in predicting lidar TI error. First, predictor variables were binned and bin-means of the TI percent error corresponding to each bin were calculated. A least-squares technique was then used to calculate a regression line between the predictor bin centers and the bin-means of the TI percent error. Sensitivity, defined as the product of the regression line slope and the standard deviation of the predictor variable, was then calculated for each predictor. The sensitivity gives the approximate change in the TI error for a change in the predictor variable that is equivalent to one standard deviation of the variable. All predictor variables had sensitivity values over 0.5, which indicates a significant relationship between all the predictor variables and the TI error, according to Annex L of IEC 61400-12-1 .~~

**3.6   Comparison to previous methods**

 In contrast to the methods discussed in Sect. 2.2, L-TERRA uses only information that is available from a standard vertically profiling lidar. The physics-based corrections in L-TERRA require only data from the lidar itself, while the machine learning module in L-TERRA can be trained using either cup or sonic anemometer data. The majority of the ~~predictor variables are related to the atmosphere, which is a highly synergistic system, it is likely that one or more of the variables are correlated to one another. Thus, a correlation matrix was calculated for the potential predictor variables. For pairs of variables with a correlation coefficient of over 0.5, the predictor with a lower sensitivity value was removed from the list of potential predictor variables. The final predictor variables were as follows: TI from the physics-based corrections, $\alpha$, SNR, $\sigma_w^2$ ($w$ velocity variance), spectral broadening, instrument internal temperature, and pitch of the lidar.~~

**3.7**

~~Reduction in TI error was related to reduction in turbine power prediction error through the use of a power prediction model. Simulations used in were again used to develop a power prediction model for the 1.5MW WindPACT turbine . First, 3-D flow fields with varying degrees of wind shear and TI were created using TurbSim . These flow fields were then used as input for the turbine simulator FAST to model the response of the WindPACT turbine to flow fields with different degrees of shear and turbulence. The 10min mean hub-height wind speed, hub-height TI, and shear parameter were extracted from the TurbSim output and the 10min mean turbine power was extracted from the FAST output. These parameters, in addition to the turbine~~

5   corrections in L-TERRA can be implemented with fewer than 20 lines of code, the models employed in L-TERRA are well-documented in the literature and simple to understand. It takes approximately 0.1 seconds to run L-TERRA
10  for a single 10min period, making it easy to implement in real time. As discussed in Sect. 5.1, a stability-dependent version of L-TERRA can be used to adapt to changing conditions and apply corrections appropriate for the current atmospheric stability regime.

**4  Data sets**

L-TERRA was tested on data from  two different locations: the Southern Great Plains ARM site in Lamont, Oklahoma
15   (Fig. 3), and an operational wind farm in the Southern Plains region of the United States. The ARM site, a field measurement site operated by the U.S. Department of Energy, contains several remote sensing and in-situ instruments (Mather and Voyles, 2013). From November 2012 to June 2013, a WC lidar owned by Lawrence Livermore National Laboratory was deployed at the ARM site approximately 100 m from a 60m tower. Gill Windmaster Pro 3-D sonic anemometers are mounted on the tower at 25 and 60 m AGL and collect velocity data at a frequency of 10 Hz.
20  The ~~BAO is a field site located approximately 25 km east of the Rocky Mountain foothills . The WC was deployed at the BAO near a 300m tower from February to April 2014. The 300m tower was instrumented with twelve 3-D sonic anemometers, with two sonics mounted on opposite booms every 50 m from 50 to 300 m AGL. RM Young sonics, owned by the University of Oklahoma, were mounted on the northwest booms of the tower, and Campbell Scientific CSAT3 sonics, owned by the National Center for Atmospheric Research, were mounted on the southeast booms.~~
25   WC was also deployed at an operational wind farm in the Southern Great Plains. (Due to a nondisclosure agreement with the wind farm, we cannot disclose the location or details of the wind farm.) The WC was located on the wind farm from November 2013 to July 2014, with a break from February to April 2014 while the WC was deployed for a different field experiment. During the wind farm deployments, the WC was sited in the same enclosure as a met tower with standard wind energy instrumentation, including a cup anemometer at the turbine hub height. For the winter
30  deployment, the WC was located near a met tower on the north end of the wind farm, and for the spring/summer deployment, the WC was moved to  a tower enclosure at the south end of the wind farm, in accordance with the dominant wind direction during each season at the wind farm. Data shown here are restricted to wind directions corresponding to turbine inflow.

Although the simulated 1.5MW WindPACT turbine used in this work has a hub height of 84 m, none of the sites were instrumented with cup or sonic anemometers at the 84m measurement height. Thus, the closest height to 80 m that contained both lidar and met tower data at each site was defined as the "hub height" for that site. This resulted in hub heights of 60, 100, and 80 m at Sample scatter plots of met tower versus lidar TI for the ARM site , BAO, and wind farm, respectively.

Histograms of the 10min mean hub-height wind speed, shear parameter, and hub-height SNR from the different sites are shown in Fig. ??. While the ARM site and Southern Plains wind farm experienced similar atmospheric conditions, conditions at the BAO site were characterized by lower values of the mean wind speed, shear parameter, and SNR. Part of this discrepancy, particularly in SNR, could be due to the different "hub heights" at each site; the SNR values shown for the BAO are from 100 m while the SNR values for the ARM site and wind farm are from 60 4a, and 80 m, respectively. As SNR is strongly tied to aerosol concentrations, which generally decrease with height, SNR is expected to be lower for measurement heights that are further from the ground . However, SNR values from the 50m height at the BAO were also generally lower than SNR values at the ARM site and wind farm. During the BAO campaign, much lower WC data availability was noted in comparison to the ARM site and wind farm campaigns. The lower data availability and SNR values at the BAO can be largely attributed to westerly flow from the direction of the Rocky Mountains, which brings cleaner air with a lower aerosol concentration to areas downwind of the mountains . also noted lower WC lidar data availability in the Colorado foothills region in comparison to a site in Iowa.

TI is shown as a function of mean wind speed and stability class for the different sites in Fig. ??. corresponding regression statistics for the raw TI are shown in Table 3. Stability classes were stratified according to the value of $\alpha$ measured by the WC between 40 and 200 m, with thresholds given in Table 2. These thresholds are loosely based on the thresholds used in Wharton and Lundquist (2012). The shear parameter $\alpha$ was used as a proxy for stability in this work, as other stability parameters such as the Richardson number and Obukhov length require temperature measurements at different heights, which are not available from a lidar.

The Normal Turbulence Model , indicated by the black lines in Fig. ??, predicts a sharp decrease in TI as $\overline{U}$ increases and the denominator in Eq. 5 becomes larger. With the exception of some outliers, this trend is largely followed for the TI and $\overline{U}$ values at the different sites. At the ARM site and wind farm, TI values measured by the lidar are close to those predicted by the NTM under neutral conditions. At these sites, low shear conditions (near-zero or negative shear parameter) tended to be associated with low wind speeds and higher TI values. Low shear conditions often occur when the atmosphere is unstable, resulting in buoyant mixing, a uniform wind speed profile, and higher amounts of turbulence . In contrast, high shear conditions (large positive shear parameter) tended to be associated with higher wind speeds and lower TI values. High shear conditions often occur when the atmosphere is stable; mixing and turbulent motions are inhibited and wind speed tends to decrease with height as frictional effects from the surface become less dominant. At the BAO, low shear conditions were often associated with very low wind speeds and high TI, similar to the other sites. However, the striation of the $\overline{U}$/TI curve by shear parameter is not as prominent at the BAO, and high TI values were often associated with large shear parameters. At this site, the magnitude of TI is likely strongly affected by the low SNR values at the site (Fig. ??) and complex terrain, in addition to the diurnal heating cycle. Thus, relations between TI, wind speed, and shear parameter are not as clear at the BAO.

Figures **??** and **??** demonstrate the large differences in atmospheric conditions and lidar data quality that can occur in different locations. Thus, the deployment of the same WC lidar at three different sites alongside met towers provides an excellent opportunity to assess the accuracy of lidar-measured TI at different locations. In addition, the large amount of lidar and met tower data collected during the experiments can be used to evaluate the effects of TI error on wind power prediction and

5 to quantify the improvement in power prediction that occurs when lidar TI estimates are improved under different atmospheric conditions.

**5 TI error and effects on power prediction**

Scatter plots of met tower versus lidar 10min mean wind speed and TI for all three sites are shown in Fig. **??**. Mean wind speeds measured by the lidar are extremely close to those measured by the met tower instruments, with regression line slopes near

10 1 and nearly all coefficient of determination ($R^2$) values greater than or equal to 0.99. There is slightly more scatter between the lidar and sonic mean wind speeds at the BAO (Fig. **??**), likely because SNR values at the BAO were lower and the lidar data quality was not as good in comparison to the other two sites. The excellent comparison of met tower and lidar mean wind speeds indicates that the WC lidar could accurately measure the mean flow at the different sites. However, large discrepancies between the TI measured by the lidar and the met tower instruments were noted at all three sites.

15 At the ARM site, $\alpha$ was strongly related to the sign of TI errors, with the WC overestimating TI under unstable conditions and underestimating TI under stable conditions (Fig. 4a). The over- and underestimation of TI was likely due to the effects of variance contamination and volume averaging, respectively. Regression line slopes increase with decreasing stability (Table 3), as in Sathe et al. (2011). In this region of the United States, the shear parameter is strongly tied to the atmospheric stability (e.g., Newman and Klein, 2014), likely because the diurnal transition of the atmospheric boundary layer largely controls the

20 wind speed profile in flat terrain (e.g., Arya, 2001). Initial TI error trends from the wind farm data set are remarkably quite similar to those found in the ARM data set (Fig. **??**c). This is not surprising, as both data sets were collected in a similar region with similar terrain and diurnal transitions.

Table 4).

In contrast, lidar TI errors at the BAO did not follow a distinct pattern according to the shear parameter (Fig. **??**). Flow in

25 this area is affected by complex terrain in addition to diurnal trends, so the shear parameter is likely not an accurate indicator of the atmospheric stability. At the BAO, nearly all the lidar TI measurements were overestimates in comparison to the sonic anemometers. As SNR values at the site were generally much lower in comparison to the ARM site (Fig. **??**), more noise was likely present in the lidar data at the BAO, resulting in TI overestimates. The lower SNR at the site also contributed to low lidar data availability and a much smaller number of data points in comparison to the other two sites.

**5 L-TERRA results**

Next, lidar and met tower TI data were used as inputs for the power prediction model, in addition to

**5.1 Application of physics-based corrections**

First, data from each site were examined individually to assess the performance of L-TERRA. For both the ARM site and the wind farm, all possible combinations of the physics-based corrections described in Section 3 were evaluated. Initially, the   model combination that produced the lowest overall TI MAE was selected as the optimal model combination for that particular site. Data were filtered to avoid mast shadowing, and 10min periods where the mean wind speed was less than 4 m s$^{-1}$  were not used to evaluate L-TERRA, as the standards outlined in IEC 61400-12-1 (International Electrotechnical Commission, 2013) restrict remote sensing classification to wind speeds between 4 and 16 m s$^{-1}$.

The optimal model combination was the same for both sites and is shown in ~~the largest power errors were found in the wind speed region just above and below the rated wind speed, where power sensitivity to turbulence is highest (Fig. ??). In this region, even small TI errors of 1%–2% can result in power errors above 2.5% of the rated power. This trend is evident at all three sites, although there are relatively fewer points in this transition region at the BAO as a result of the lower mean wind speeds experienced at~~ first row of Table 5. Several slightly different model combinations produced similar MAE values at both sites, suggesting that there may actually be multiple optimal combinations of L-TERRA at each site when the MAE is minimized. It may be useful to consider other parameters in determining the optimal model combination, such as regression line statistics or sensitivity of TI error to atmospheric stability. However, minimizing the MAE is a standard approach for determining optimal model combinations and provides a useful baseline combination for evaluating L-TERRA. Application of this initial L-TERRA model combination resulted in a modest reduction of lidar TI MAE from 1.5% to 1.4% at the ARM site and from 1.48% to 1.39% at the wind farm (not shown).

By examining the change in lidar TI after each step in L-TERRA, it was determined that some corrections decreased error under stable conditions while increasing error under unstable conditions, and vice versa. This is not surprising, as the magnitude and sign of TI errors was strongly dependent on atmospheric stability at both sites (Tables 3, 4) as a result of the different factors that affect TI error under different stability conditions. Thus, optimal model combinations were next determined separately for the ~~BAO in comparison to the other two sites (Fig. ??). The largest TI errors occur at lower wind speeds (near 75% of the rated wind speed). However, because power sensitivity to TI is low in this region of the power curve, these large TI errors did not often translate to large errors in predicted power. For wind speeds above the rated wind speed, power error increases steadily with increasing TI error, but most power errors are below 0.5% of the rated power.~~three different bulk stability classes to form a stability-dependent version of L-TERRA (L-TERRA-S). Optimal model combinations were very similar for both sites and are shown in Table 5.

 For all three stability classes, a spike filter was the optimal noise

removal technique. Only the model chain for stable conditions ~~). At all three sites, power errors are negligible for wind speeds near 75% of the rated wind speed and are generally smaller than 0.5% for wind speeds around 125% of the rated wind speed. The largest errors , as also shown in Fig. **??**, occur for wind speeds near rated. (Note the small number of colored boxes in Fig. **??** for wind speeds near and above the rated wind speed. Colored boxes are only shown for bins with three or more data points, and wind speeds at the BAO were generally quite low, as previously discussed. ) For the ARM site and the wind farm, the sign of the errors changes when moving from stable to unstable conditions , with power overestimates occurringand power underestimates occurring under unstable conditions. As the WC typically underestimates TI under stable conditions (Fig. **??**), power predictions made with these TI estimates underestimate the effects of TI on power (Fig. 1). Thus, power predictions made under stable conditions with the WCTI values are overestimates oftrue power. In contrast, the WC overestimates TI under unstable conditions (Fig. **??**) and thus overestimates the effect of TI on power. In the region near the rated wind speed, overestimating the TI results in predicting a lower amount of power than what is truly produced (Fig. 1). At all three sites, the largest power errors tend to occur under low-shear, high TI conditions, which typically correspond to unstable conditions. This is not surprising, as WC TI estimatesunder unstable conditionshad the largest errors in comparison to met tower measurements, as evidenced by the large slopes~~ VAD technique is typically applied once per full scan to derive the three-dimensional wind vector. For the WC, this results in an output data frequency of 0.25 Hz for the VAD technique. The lower temporal resolution of the VAD technique likely served to artificially reduce some of the effects of variance contamination, as smaller scales of turbulence were not measured.

[revised manuscript text omitted]

Sample plots showing the response of TI percent error to different variables at the ARM site are shown in Fig.

15 5. Raw WC TI error was extremely sensitive to the four variables depicted in Fig. 5, with larger TI percent errors for lower wind speeds and SNR values, ~~low-TI conditions (stable). Although L-TERRA improves these TI estimates, bringing them very close to the one-to-one line (Figs. **??, ??**), these periods are still associated with large power percent errors . This likely occurs because at both of these sites, low-TI, high-shear conditions often correspond to higher wind speeds near the rated wind speed of the 1.5MW WindPACT turbine (Figs. **??, ??**), where turbulence sensitivity is~~

20 ~~highest (Fig. **??**). Other large power errors are associated with unstable and neutral conditions with higher TI values. Although the largest TI errors occur under low-shear, high-TI conditions (Figs. 4a, **??**c, **??, ??**), these large TI errors do not often result in large power errors, as they are usually associated with low wind speeds at both the ARM site and the wind farm (Figs. **??, ??**), where turbulence sensitivity is low (Fig. **??**). Several of the neutral and unstable TI estimates that are associated with large power errors are initially located above the one-to-one line in Figs. **??** and **??**; these overestimates are likely a result of~~

[revised manuscript text omitted]
 for FAST simulations. Only data where the shear exponent is approximately equal to 0.2 are shown. Sensitivity was approximated as the regression line slope for power versus TI in different wind speed bins.

[Figure]

**Figure 2.** Flowchart depicting different methods for correcting TI with L-TERRA. Starting and ending points are indicated by blue-outlined ovals and modules are indicated by red-outlined diamonds.

[Figure]

**Figure 3.** Inset: Google Earth image of the state of Oklahoma. Location of Southern Great Plains ARM site is denoted by white marker. Larger figure: Google Earth image of the central facility of the Southern Great Plains ARM site (outlined in red box) with overlaid elevation contours in feet. Elevation map is from the United States Geological Survey and uses contour intervals of approximately 10 feet (3.05 m). Locations of WC lidar and 60m tower are indicated by white markers.

[Figure]

**Figure 4.** Scatter plots of met tower vs. WC TI for data from 60m measurement height at the ARM site a) before and b) after L-TERRA-S has been applied. One-to-one line and regression lines are shown for reference and regression line statistics are shown in figure legends.

[Figure]

**Figure 5.** Percent difference between WC and sonic TI for the ARM site as a function of a) shear parameter b) raw WC TI c) mean wind speed and d) SNR. Differences are shown both before (red circles) and after (blue circles) L-TERRA-S has been applied. Solid circles correspond to averages of binned data and solid lines correspond to regression line fits to bin-means, following the procedure in Annex L of IEC 61400-12-1, Draft Edition (International Electrotechnical Commission, 2013).

[Figure]

(a) Random Forest 1 (six input variables)    (b) Random Forest 2 (four input variables)

**Figure 6.** Scatter plots of met tower vs. WC TI for data from 60m measurement height at the ARM site a) after application of the first random forest described in the text and b) after application of the second random forest. One-to-one line and regression lines are shown for reference.

---

## Referee Report (RR1)

In the second version of this manuscript, the first 5 pages are introductory material.

The shear parameter does not appear to be an adequate measure of stability and it could be that this is why the scatter and errors remain large. The shear parameter includes roughness and terrain effects and is likely direction sensitive. If the authors really insist that they do not have to compare with stability determined from sonic data (which they have available) then at least they should call this a shear parameter and not stability, or show that it is equivalent.

It is not clear which of the steps in sections 3.1-3.5 have been applied and whether they are needed. What are the physics-based corrections in L-TERRA? Maybe the errors in the large shear case are because the exponent used is wrong? It is unfortunate that the r2 values are so little impacted. The MAE improvement of 0.26 is very difficult to evaluate. Does it mean the difference between the met tower TI (height?) and the WC TI was previously larger than 1.50 so for example it could be that WCTI was 11.5% while the met tower TI is 10% and with TERRA it is now predicted to be 11.26%? Is that a correct interpretation? At the wind farm site the improvement in MAE appears to be 0.28. Are these really significant as stated in the abstract when there is very little or no change in the slope or r2 values?

Section 3.6 is called comparison to previous methods but there is no quantitative comparison. Nor is there a qualitative evaluation of which of the preprocessing steps are necessary and have utility. It is unfortunate because a quantitative comparison here would add value, even if the overall results show rather small improvements from using L-TERRA.

With a more quantitative approach to the processing there could be something here. But based on these results and with the lack of any physical detail of the model, or processing detail that indicates the value of the steps it is unfortunately not a very compelling analysis.

---

## Referee Report (RR2)

**Review of**

**"An Error Reduction Algorithm to Inprove Lidar Turbulence Estimates for Wind Energy"**

Jennifer F. Newman and Andrew Clifton

Submitted to Wind Energy Science- October 2016

General comments:

The paper present the L-TERRA algorithm developed for correcting ground based profiler lidar measured turbulence intensity. Correcting here implies retrieving the same TI as that measured by a sonic anemometer. The approach is interesting since there is an incontestable need for a "practical" way to use TI measured by Doppler lidars.

The paper has been very much improved since the first submission (version from July 2016). It is now very focused and comprehensive. The comments from the first review have been well taken into account.

The analysis has been further detailed in several aspects:

1. Intermediate results from the L-TERRA-S algorithm showed a better agreement on average of lidar TI with sonic/cup TI with the physic based correction (correction for noise, spatial averaging and variance contamination), before applying a machine learning process. The regression slope was closer to one but the scatter was slightly increased.
2. The process was tested on subsets of the datasets corresponding to different atmospheric stability classes and showed that different combinations of corrections were required for different stability classes (stable conditions require extrapolation of the power spectrum to compensate for the volume averaging effect, unstable conditions require correction for the cross contamination). Those results are coherent with previous studies on turbulence measured by Doppler lidars.
3. Two different machine learning process were tested out and it was shown that it did not improve the results obtained with the physics-based corrections alone.

The paper is well written with a good introduction, a comprehensive literature review, clear results and clear conclusions. The abstract is also very comprehensive.

Below are minor detailed comments.

Detailed comments:

P2, l16 to 22 and Figure 1: This is absolutely correct but this could be made shorter as it is done for resource assessment. The effect of turbulence on power curve is fairly well known, it is therefore not necessary to discuss and illustrate it here. A statement that turbulence intensity measurement is necessary

for power performance measurement with some relevant references (e.g. Malcolm and Hansen, 2006) would be enough.

P7, l17:

1. could you give more details on the "constant temporal spacing" used here? Is it 1s and 4s? Have the sonic and cup data been down sampled to have identical frequency?
2. Has any data filtering been applied e.g. on SNR?

P8, section 3.2:

 Would it be possible to have a brief explanation of the spike filter and the Lenschow methods mentioned here so that a reader who does not know about them can get an idea of what it is about without having to read those papers?

P8, l18: the word "evaluated" is confusing here as you have actually only applied one of them (the second one: modeling and extrapolation of the spectrum). Please try to make it clear throughout sections 3 and 5 that you have only evaluated methods applied to u,v,w (left side of the flow chart in Fig 2) and not those applied to Vrad). I suggest to make it is clear here that you are describing all methods but have not tested them all. Suggestion for re-wording: "Two methods can be used to mitigate…".

P10 and l12: could you provide some information about the flatness/complexity of the site (e.g. max elevation difference within a couple of km around the WC)

P10, l11 and l16: was the WC configured to measure at the same heights as the cups/sonics used as reference? E.g. 60m at ARM site

P10, l19: would it be possible to indicate how far from the turbines the WC was located (any effect from induction within conical scan)?

P10, l20: could you also provide some indication of the comparison of the mean wind speed between the WC and the reference instrument? If the mean wind speed comparison is not nearly "perfect" (1:1 slope and very low scatter) there is little chance that the TI compares well.

P11, l9-15:  a more quantitative description of the results would be relevant:

1. What was the MAE range (maximum) overall tested combinations? Has the MAE been increased in some cases compared to the MAE without correction? This would show that wrong choices of correction or wrong combination would be worse than no correction at all.
2. What do you mean by "similar MAE values" (line 10)? Could you give a range?
3. How many other methods were in that range?

P12, l3: interesting results with the different stability classes. Could the scatter be due to the proxy used to define the classes? I mean could the scatter be due to data for which the stability class is not what is approximated by the shear exponent and in those cases the correction applied is not optimal?

I think it would be very relevant to discuss this a bit further in the paper as in the end it is mitigating a bit the "practical" approach of the method since, from the results presented now, it seems that to apply an

effective correction to the lidar TI, we first need to identify the stability class for each 10 minute data. Quantifying the stability class from solely lidar information is a challenge.

Table 3:

1. Could you please indicate the number of data for each data set or subset?
2. Indicate that the results are for the ARM site in the caption

Table 5:

The wind speed column is not clear. It can be confused with the "Wind speed" decision box of the flow chart in Fig 2. I suggest to indicate "1 sec" and "4 sec" or "1Hz" and "0.25Hz2 instead of "raw" and "VAD".

Figure 2:

1. What do you mean by "raw WC data" in top box? Unfiltered data? Radial wind speeds?
2. The option for not appliying a correction ( e.g. for volume averaging) does not appear clearly as an option. Perhaps it is represented by the direct arrow between "Volume averaging?" and "Variance contamination?" but then there should also be arrows between "Spectral fit 1/2" and "Variance contamination?" to be consistent. And that shoud be consistent through the whole flow chart.

---

## Author Response (AR2)

Response to Reviewer 1

Dear Reviewer,

Thank you for your review of the revised manuscript. In response to your comments, we have further revised the manuscript to include a discussion of stability parameters, including a comparison of lidar shear exponent and gradient Richardson number at the ARM site and a discussion of the potential impacts of stability misclassification on L-TERRA performance. We have also added a discussion on the impact of the different physics-based correction modules on the TI error. Responses to detailed comments are listed below. Please note that statistics associated with L-TERRA changed slightly for this version of the manuscript due to a recent modification of our spike filter routine.
* * *
*In the second version of this manuscript, the first 5 pages are introductory material.*

**Response:** We acknowledge that this is a substantial amount of introductory material, but we feel it is helpful for informing the reader about the most significant sources of lidar TI error and current techniques for reducing lidar TI error.

*The shear parameter does not appear to be an adequate measure of stability and it could be that this is why the scatter and errors remain large. The shear parameter includes roughness and terrain effects and is likely direction sensitive. If the authors really insist that they do not have to compare with stability determined from sonic data (which they have available) then at least they should call this a shear parameter and not stability, or show that it is equivalent.*

**Response:** We have now included a discussion of stability classification in the revised manuscript. Section 4.2 was added to explain our justification for using the lidar shear exponent as a proxy for stability and to compare shear exponents measured at the ARM site to corresponding values of the gradient Richardson number calculated from tower data at the site. Opposing stability classifications were only made in 5% of the time periods we examined, with most stability misclassifications being made in near-neutral conditions, as defined by either the shear exponent or the Richardson number.

In Section 5.1.3, we also discuss the potential impacts of stability misclassification on performance of L-TERRA. We acknowledge that this is a source of uncertainty and suggest that it may be useful to use additional parameters to classify stability and to treat near-neutral cases separately from cases that are clearly stable or unstable.

*It is not clear which of the steps in sections 3.1-3.5 have been applied and whether they are needed. What are the physics-based corrections in L-TERRA?*

**Response:** Section 3 has now been split into three subsections: pre-processing, physics-based corrections, and machine learning, to clarify what the physics-based corrections in L-TERRA include. In Section 5.1.2, we have now included a discussion of the impact of

the different physics-based corrections on the lidar TI (p. 15, Lines 4-17, Figure 5).

*Maybe the errors in the large shear case are because the exponent used is wrong?*

**Response:** We have added a discussion on the impact of incorrect stability classifications on L-TERRA performance (Section 5.1.3). We do acknowledge that incorrect stability classifications may have led to some of the large TI errors but do not believe this is the only explanation for the large errors.

*It is unfortunate that the r2 values are so little impacted. The MAE improvement of 0.26 is very difficult to evaluate. Does it mean the difference between the met tower TI (height?) and the WC TI was previously larger than 1.50 so for example it could be that WCTI was 11.5% while the met tower TI is 10% and with TERRA it is now predicted to be 11.26%? Is that a correct interpretation? At the wind farm site the improvement in MAE appears to be 0.28. Are these really significant as stated in the abstract when there is very little or no change in the slope or r2 values?*

**Response:** We agree that the changes in $R^2$ and MAE values are not very significant, although the large improvement in regression line slopes and decrease in sensitivity to stability/shear suggest that we are on the right track with the corrections we are applying. We believe that with some refinements to the corrections, we can improve the $R^2$ and MAE values.

The improved MAE of 1.24% implies that the mean absolute error after L-TERRA has been applied is 1.24%, but some errors are larger than this value while other errors are smaller. At the ARM site, the application of L-TERRA reduced some errors by up to 2-4% (so, for example, if the original WC TI was 15% and the sonic TI was 10%, then the WC TI after L-TERRA was applied could be 11%).

We have now revised the abstract to highlight the improvement in regression line slopes and decrease in sensitivity to stability, rather than the reduction in MAE, which, as you mentioned, is not extremely significant.

*Section 3.6 is called comparison to previous methods but there is no quantitative comparison. Nor is there a qualitative evaluation of which of the preprocessing steps are necessary and have utility. It is unfortunate because a quantitative comparison here would add value, even if the overall results show rather small improvements from using L-TERRA.*

**Response:** We agree that a quantitative comparison of our method to previous methods would be helpful for putting our method in the context of previous work. However, most of the methods we cited in our paper require either the use of scanning lidars (e.g., Krishnamurthy et al. 2011, Fuertes et al. 2014) or the use of Doppler spectra (e.g., Mann et al. 2010, Branlard et al. 2013), which are not typically available from a commercial lidar. This precluded us from applying these corrections to our own data. Some papers only analyze short time periods of data and do not provide statistics for their corrections

(e.g., Sjöholm et al. 2009, Krishnamurthy et al. 2011, Fuertes et al. 2014), while others only model the expected variance measured by a lidar but do not apply corrections (e.g., Sathe et al. 2011), making it difficult to compare our improvements in TI to improvements in variance achieved by other methods.

*With a more quantitative approach to the processing there could be something here. But based on these results and with the lack of any physical detail of the model, or processing detail that indicates the value of the steps it is unfortunately not a very compelling analysis.*

**Response:** We appreciate your comments and hope we have addressed your concerns by adding a discussion about stability parameters, clarifying several steps in the model, and discussing the importance of each step in L-TERRA. We welcome any further suggestions to improve the manuscript.

References

Branlard, E., Pedersen, A. T., Mann, J., Angelou, N., Fischer, A., Mikkelsen, T., Harris, M., Slinger, C., and Montes, B. F.: Retrieving wind statistics from average spectrum of continuous-wave lidar, Atmos. Meas. Tech., 6, 1673–1683, doi:10.5194/amt-6-1673-2013, 2013.

Fuertes, F. C., Iungo, G. V., and Porté-Agel, F.: 3D turbulence measurements using three synchronous wind lidars: Validation against sonic anemometry, J. Atmos. Oceanic Technol., 31, 1549–1556, doi:10.1175/JTECH-D-13-00206.1, 2014.

Krishnamurthy, R., Calhoun, R., Billings, B., and Doyle, J.: Wind turbulence estimates in a valley by coherent Doppler lidar, Meteorological Applications, 18, 361–371, doi:10.1002/met.263, 2011.

Mann, J., Peña, A., Bingöl, F., Wagner, R., and Courtney, M. S.: Lidar scanning of momentum flux in and above the atmospheric surface layer, J. Atmos. Oceanic Technol., 27, 959–976, doi:10.1175/2010JTECHA1389.1, 2010.

Sathe, A., Mann, J., Gottschall, J., and Courtney, M. S.: Can wind lidars measure turbulence?, J. Atmos. Oceanic Technol., 28, 853–868, doi:10.1175/JTECH-D-10-05004.1, 2011.

Sjöholm, M., Mikkelsen, T., Mann, J., Enevoldsen, K., and Courtney, M.: Spatial averaging-effects on turbulence measured by a continuous wave coherent lidar, Meteor. Z., 18, 281–287, doi:10.1127/0941-2948/2009/0379, 2009.

Dear Rozenn,

We sincerely thank you for your detailed review of the revised manuscript. In response to your comments, we have further revised the manuscript to clarify the data processing techniques and evaluation methods used to develop L-TERRA. We have also included a discussion of stability classification based on the shear exponent measured by the lidar. Responses to detailed comments are listed below. Please note that statistics associated with L-TERRA changed slightly for this version of the manuscript due to a recent modification of our spike filter routine.
* * *
*P2, l16 to 22 and Figure 1: This is absolutely correct but this could be made shorter as it is done for resource assessment. The effect of turbulence on power curve is fairly well known, it is therefore not necessary to discuss and illustrate it here. A statement that turbulence intensity measurement is necessary for power performance measurement with some relevant references (e.g. Malcolm and Hansen, 2006) would be enough.*

**Response:** Figure 1 has now been removed and references have instead been given for the effects of turbulence on the power curve.

*P7, l17:*
*1. could you give more details on the "constant temporal spacing" used here? Is it 1s and 4s? Have the sonic and cup data been down sampled to have identical frequency?*

**Response:** We have now clarified that the "constant temporal spacing" is 1 Hz for the 1-s data and 0.25 Hz for the 4-s data.

The sonic data were available at 10 Hz and we did try downsampling the data to 1 Hz and 0.25 Hz. TI values were nearly identical for the 10 Hz and 1 Hz data, with a mean absolute error of less than 0.1%.  Slightly larger discrepancies were apparent between the 10 Hz and downsampled 0.25 Hz data, although the mean absolute error was still quite low at approximately 0.5%. Thus, we felt confident in using the original 10 Hz sonic data in our analysis, as factors such as volume averaging and variance contamination appear to have a much larger effect on lidar TI errors than temporal resolution.

Only ten-minute averages of wind speed and standard deviation were available for the cup data, although we believe the cup data were originally collected at a frequency of 1 Hz.

*2. Has any data filtering been applied e.g. on SNR?*

**Response:** The WindCube model used in this work automatically removes data that were associated with SNR values below -22 dB, so no additional filtering was applied on SNR.

*P8, section 3.2:*
*Would it be possible to have a brief explanation of the spike filter and the Lenschow methods mentioned here so that a reader who does not know about them can get an idea of what it is about without having to read those papers?*

**Response:** Brief explanations of the two noise removal techniques have now been added to the manuscript.

*P8, l18: the word "evaluated" is confusing here as you have actually only applied one of them (the second one: modeling and extrapolation of the spectrum). Please try to make it clear throughout sections 3 and 5 that you have only evaluated methods applied to u,v,w (left side of the flow chart in Fig 2) and not those applied to Vrad). I suggest to make it is clear here that you are describing all methods but have not tested them all. Suggestion for re-wording: "Two methods can be used to mitigate…".*

**Response:** We have now made it clear in Sections 3 and 5 that only the u, v, and w methods were evaluated, though all methods are described. We have also changed "Two methods were evaluated…" to "Two methods were considered…"

*P10 and l12: could you provide some information about the flatness/complexity of the site (e.g. max elevation difference within a couple of km around the WC)*

**Response:** This information has now been added to the beginning of Section 4.

*P10, l11 and l16: was the WC configured to measure at the same heights as the cups/sonics used as reference? E.g. 60m at ARM site*

**Response:** Yes, the WC was configured to measure at the same height as the reference instruments (although, this height of course corresponds to the center of the probe volume). This has now been clarified at the beginning of Section 4.

*P10, l19: would it be possible to indicate how far from the turbines the WC was located (any effect from induction within conical scan)?*

**Response:** The closest turbines were located approximately 1.7$D$ from the edge of the WC scanning circle at 80 m, so we do not expect induction to have a large effect on velocities within the scanning circle. However, this is a good factor to consider for lidar deployments on wind farms.

*P10, l20: could you also provide some indication of the comparison of the mean wind speed between the WC and the reference instrument? If the mean wind speed comparison is not nearly "perfect" (1:1 slope and very low scatter) there is little chance that the TI compares well.*

**Response:** The mean wind speeds between the WC and the reference instruments did compare very well at both sites (slope $\approx 1$, $R^2 \approx 0.99$). This has now been mentioned in the manuscript in the first paragraph of Section 4.3.

*P11, l9-15: a more quantitative description of the results would be relevant:*
*1. What was the MAE range (maximum) overall tested combinations? Has the MAE been increased in some cases compared to the MAE without correction? This would show that wrong choices of correction or wrong combination would be worse than no correction at all.*

**Response:** This information has now been added to Section 5.1.1 of the manuscript. The MAE did increase in comparison to the original MAE for some model combinations, and some possible explanations for why this occurred are given in the second paragraph of Section 5.1.1.

*2. What do you mean by "similar MAE values" (line 10)? Could you give a range?*

*3. How many other methods were in that range?*

**Response:** These questions have now been addressed in the revised version of the manuscript. It was determined that very similar MAE values were obtained for model combinations where the noise removal technique had a very small impact on the resulting TI. Thus, model combinations that were the same except for the noise removal technique produced very similar MAE values.

*P12, l3: interesting results with the different stability classes. Could the scatter be due to the proxy used to define the classes? I mean could the scatter be due to data for which the stability class is not what is approximated by the shear exponent and in those cases the correction applied is not optimal?*
*I think it would be very relevant to discuss this a bit further in the paper as in the end it is mitigating a bit the "practical" approach of the method since, from the results presented now, it seems that to apply an effective correction to the lidar TI, we first need to identify the stability class for each 10 minute data. Quantifying the stability class from solely lidar information is a challenge.*

**Response:** This is a very valid point and we have now addressed the issue of stability classification in the revised manuscript. Section 4.2 was added to discuss our justification for using the shear exponent to classify stability and to compare the shear exponents measured at the ARM site to the corresponding gradient Richardson number values calculated from met tower data. We determined that opposing classifications were only made for 5% of the periods we examined and that many misclassifications occurred near neutral conditions. This is not surprising, as a small error in lidar wind speed could change the shear exponent enough to change the stability classification from neutral to near-stable or near-unstable, or vice versa. Likewise, small errors in temperature measured by the probes on the met tower could change the sign of the temperature gradient and thus the stability classification made by the Richardson number.

We also now discuss the impact of stability misclassifications on the performance of L-TERRA in Section 5.1.3. We acknowledge that stability misclassification is a source of error and that it may be useful to employ additional lidar parameters to classify stability and to treat near-neutral cases differently than cases that are clearly stable or unstable. However, we do not feel that stability misclassification is the largest source of error for L-TERRA, as it accounted for less than 10% of the large TI errors after L-TERRA was applied.

*Table 3:*
*1. Could you please indicate the number of data for each data set or subset?*
*2. Indicate that the results are for the ARM site in the caption*

**Response:** These changes have been made to Table 3. The number of data points for each data subset has also been added to Table 4.

*Table 5:*
*The wind speed column is not clear. It can be confused with the "Wind speed" decision box of the flow chart in Fig 2. I suggest to indicate "1 sec" and "4 sec" or "1Hz" and "0.25Hz2 instead of "raw" and "VAD".*

**Response:** This column has now been clarified by renaming the column to "Wind Speed Frequency" and changing the "Raw" and "VAD" boxes in the column to "1 Hz" and "0.25 Hz", as suggested.

*Figure 2:*
*1. What do you mean by "raw WC data" in top box? Unfiltered data? Radial wind speeds?*

*2. The option for not applying a correction (e.g. for volume averaging) does not appear clearly as an option. Perhaps it is represented by the direct arrow between "Volume averaging?" and "Variance contamination?" but then there should also be arrows between "Spectral fit 1/2" and "Variance contamination?" to be consistent. And that should be consistent through the whole flow chart.*

**Response:** The flowchart has now been modified for clarification. "Raw WC Data" has been changed to "High-Frequency Lidar Output Files", indicating that L-TERRA starts with the data from the high-frequency output files from the lidar (i.e., the 1 Hz files from the WC). Boxes have been added for the noise removal, volume averaging, variance contamination, and machine learning decision points to indicate that it is also an option to not apply a correction for that particular module. This information has also been clarified in the text.

[revised manuscript text omitted]